# PATHFORMER: MULTI-SCALE TRANSFORMERS WITH ADAPTIVE PATHWAYS FOR TIME SERIES FORECASTING

**Peng Chen**[1*]**, Yingying Zhang**[2]**, Yunyao Cheng**[3]**, Yang Shu**[1†]**, Yihang Wang**[1*]**,**
**Qingsong Wen**[2]**, Bin Yang**[1]**, Chenjuan Guo**[1]
[1]East China Normal University, [2]Alibaba Group, [3]Aalborg University
{pchen,yhwang}@stu.ecnu.edu.cn, congrong.zyy@alibaba-inc.com
{yshu,cjguo,byang}@dase.ecnu.edu.cn, yunyaoc@cs.aau.dk
qingsongedu@gmail.com

## ABSTRACT

Transformers for time series forecasting mainly model time series from limited or fixed scales, making it challenging to capture different characteristics spanning various scales. We propose Pathformer, a multi-scale Transformer with adaptive pathways. It integrates both temporal resolution and temporal distance for multi-scale modeling. Multi-scale division divides the time series into different temporal resolutions using patches of various sizes. Based on the division of each scale, dual attention is performed over these patches to capture global correlations and local details as temporal dependencies. We further enrich the multi-scale Transformer with adaptive pathways, which adaptively adjust the multi-scale modeling process based on the varying temporal dynamics of the input, improving the accuracy and generalization of Pathformer. Extensive experiments on eleven real-world datasets demonstrate that Pathformer not only achieves state-of-the-art performance by surpassing all current models but also exhibits stronger generalization abilities under various transfer scenarios. The code is made available at https://github.com/decisionintelligence/pathformer.

## 1 INTRODUCTION

Time series forecasting is an essential function for various industries, such as energy, finance, traffic, logistics, and cloud computing (Chen et al., 2012; Cirstea et al., 2022b; Ma et al., 2014; Zhu et al., 2023; Pan et al., 2023; Pedersen et al., 2020), and is also a foundational building block for other time series analytics, e.g., outlier detection Campos et al. (2022); Kieu et al. (2022b). Motivated by its widespread application in sequence modeling and impressive success in various fields such as CV and NLP (Dosovitskiy et al., 2021; Brown et al., 2020), Transformer (Vaswani et al., 2017) receives emerging attention in time series (Wen et al., 2023; Wu et al., 2021; Chen et al., 2022; Liu et al., 2022c). Despite the growing performance, recent works have started to challenge the existing designs of Transformers for time series forecasting by proposing simpler linear models with better performance (Zeng et al., 2023). While the capabilities of Transformers are still promising in time series forecasting (Nie et al., 2023), it calls for better designs and adaptations to fulfill its potential.

Real-world time series exhibit diverse variations and fluctuations at different temporal scales. For instance, the utilization of CPU, GPU, and memory resources in cloud computing reveals unique temporal patterns spanning daily, monthly, and seasonal scales Pan et al. (2023). This calls for multi-scale modeling (Mozer, 1991; Ferreira et al., 2006) for time series forecasting, which extracts temporal features and dependencies from various scales of temporal intervals. There are two aspects to consider for multiple scales in time series: temporal resolution and temporal distance. *Temporal resolution* corresponds to how we view the time series in the model and determines the length of each temporal patch or unit considered for modeling. In Figure 1, the same time series can be divided

---

[*]Part of the work was done during the internship at Alibaba Group.
[†]Corresponding author

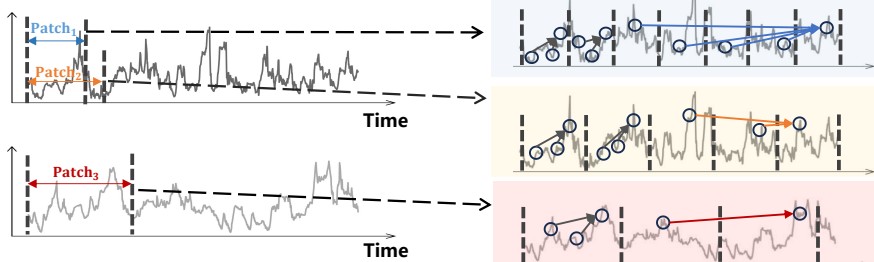

Figure 1: Left: Time series are divided into patches of varying sizes as *temporal resolution*. The intervals in blue, orange, and red represent different patch sizes. Right: Local details (black arrows) and global correlations (color arrows) are modeled through different *temporal distances*.

into small patches (blue) or large ones (yellow), leading to fine-grained or coarse-grained temporal characteristics. *Temporal distance* corresponds to how we explicitly model temporal dependencies and determines the distances between the time steps considered for temporal modeling. In Figure 1, the black arrows model the relations between nearby time steps, forming local details, while the colored arrows model time steps across long ranges, forming global correlations.

To further explore the capability of extracting correlations in Transformers for time series forecasting, in this paper, we focus on the aspect of enhancing multi-scale modeling with the Transformer architecture. Two main challenges limit the effective multi-scale modeling in Transformers. The first challenge is the *incompleteness of multi-scale modeling*. Viewing the data from different temporal resolutions implicitly influences the scale of the subsequent modeling process (Shabani et al., 2023). However, simply changing temporal resolutions cannot emphasize temporal dependencies in various ranges explicitly and efficiently. On the contrary, considering different temporal distances enables modeling dependencies from different ranges, such as global and local correlations (Li et al., 2019). However, the exact temporal distances of global and local intervals are influenced by the division of data, which is incomplete from a single view of temporal resolution. The second challenge is the *fixed multi-scale modeling process*. Although multi-scale modeling reaches a more complete understanding of time series, different series prefer different scales depending on their specific temporal characteristics and dynamics. For example, comparing the two series in Figure 1, the series above shows rapid fluctuations, which may imply more attention to fine-grained and short-term characteristics. The series below, on the contrary, may need more focus on coarse-grained and long-term modeling. The fixed multi-scale modeling for all data hinders the grasp of critical patterns of each time series, and manually tuning the optimal scales for a dataset or each time series is time-consuming or intractable. Solving these two challenges calls for *adaptive multi-scale modeling*, which adaptively models the current data from certain multiple scales.

Inspired by the above understanding of multi-scale modeling, we propose Multi-scale Transformers with Adaptive Pathways (**Pathformer**) for time series forecasting. To enable the ability of more complete multi-scale modeling, we propose a multi-scale Transformer block unifying multi-scale temporal resolution and temporal distance. Multi-scale division is proposed to divide the time series into patches of different sizes, forming views of diverse temporal resolutions. Based on each size of divided patches, dual attention encompassing inter-patch and intra-patch attention is proposed to capture temporal dependencies, with inter-patch attention capturing global correlations across patches and intra-patch attention capturing local details within individual patches. We further propose adaptive pathways to activate the multi-scale modeling capability and endow it with adaptive modeling characteristics. At each layer of the model, a multi-scale router adaptively selects specific sizes of patch division and the subsequent dual attention in the Transformer based on the input data, which controls the extraction of multi-scale characteristics. We equip the router with trend and seasonality decomposition to enhance its ability to grasp the temporal dynamics. The router works with an aggregator to adaptively combine multi-scale characteristics through weighted aggregation. The layer-by-layer routing and aggregation form the adaptive pathways of multi-scale modeling throughout the Transformer. To the best of our knowledge, this is the first study that introduces adaptive multi-scale modeling for time series forecasting. Specifically, we make the following contributions:

- We propose a multi-scale Transformer architecture. It integrates the two perspectives of temporal resolution and temporal distance and equips the model with the capacity of a more complete multi-scale time series modeling.

- We further propose adaptive pathways within multi-scale Transformers. The multi-scale router with temporal decomposition works together with the aggregator to adaptively extract and aggregate multi-scale characteristics based on the temporal dynamics of input data, realizing adaptive multi-scale modeling for time series.

- We conduct extensive experiments on different real-world datasets and achieve state-of-the-art prediction accuracy. Moreover, we perform transfer learning experiments across datasets to validate the strong generalization of the model.

## 2 RELATED WORK

**Time Series Forecasting.** Time series forecasting predicts future observations based on historical observations. Statistical modeling methods based on exponential smoothing and its different flavors serve as a reliable workhorse for time series forecasting (Hyndman & Khandakar, 2008; Li et al., 2022a). Among deep learning methods, GNNs model spatial dependency for correlated time series forecasting (Jin et al., 2023a; Wu et al., 2020; Zhao et al., 2024; Cheng et al., 2024; Miao et al., 2024; Cirstea et al., 2021). RNNs model the temporal dependency (Chung et al., 2014; Kieu et al., 2022a; Wen et al., 2017; Cirstea et al., 2019). DeepAR (Rangapuram et al., 2018) uses RNNs and autoregressive methods to predict future short-term series. CNN models use the temporal convolution to extract the sub-series features (Sen et al., 2019; Liu et al., 2022a; Wang et al., 2023). TimesNet (Wu et al., 2023a) transforms the original one-dimensional time series into a two-dimensional space and captures multi-period features through convolution. LLM-based methods also show effective performance in this field (Jin et al., 2023b; Zhou et al., 2023). Additionally, some methods are incorporating neural architecture search to discover optimal architectures(Wu et al., 2022; 2023b).

Transformer models have recently received emerging attention in time series forecasting (Wen et al., 2023). Informer (Zhou et al., 2021) proposes prob-sparse self-attention to select important keys, Triformer (Cirstea et al., 2022a) employs a triangular architecture, which manages to reduce the complexity. Autoformer (Wu et al., 2021) proposes auto-correlation mechanisms to replace self-attention for modeling temporal dynamics. FEDformer (Zhou et al., 2022) utilizes fourier transformation from the perspective of frequency to model temporal dynamics. However, researchers have raised concerns about the effectiveness of Transformers for time series forecasting, as simple linear models prove to be effective or even outperform previous Transformers (Li et al., 2022a; Challu et al., 2023; Zeng et al., 2023). Meanwhile, PatchTST (Nie et al., 2023) employs patching and channel independence with Transformers to effectively enhance the performance, showing that the Transformer architecture still has its potential with proper adaptation in time series forecasting.

**Multi-scale Modeling for Time Series.** Modeling multi-scale characteristics proves to be effective for correlation learning and feature extraction in the fields such as computer vision (Wang et al., 2021; Li et al., 2022b; Wang et al., 2022b) and multi-modal learning (Hu et al., 2020; Wang et al., 2022a), which is relatively less explored in time series forecasting. N-HiTS (Challu et al., 2023) employs multi-rate data sampling and hierarchical interpolation to model features of different resolutions. Pyraformer (Liu et al., 2022b) introduces a pyramid attention to extract features at different temporal resolutions. Scaleformer (Shabani et al., 2023) proposes a multi-scale framework, and the need to allocate a predictive model at different temporal resolutions results in higher model complexity. Different from these methods, which use fixed scales and cannot adaptively change the multi-scale modeling for different time series, we propose a multi-scale Transformer with adaptive pathways that adaptively model multi-scale characteristics based on diverse temporal dynamics.

## 3 METHODOLOGY

To effectively capture multi-scale characteristics, we propose multi-scale Transformers with adaptive pathways (named **Pathformer**). As depicted in Figure 2, the whole forecasting network is composed of Instance Norm, stacking of Adaptive Multi-Scale Blocks (**AMS Blocks**), and Predictor. Instance Norm (Kim et al., 2022) is a normalization technique employed to address the distribution shift between training and testing data. Predictor is a fully connected neural network, proposed due to its applicability to forecasting for long sequences (Zeng et al., 2023; Das et al., 2023).

The core of our design is the AMS Block for adaptive modeling of multi-scale characteristics, which consists of the multi-scale Transformer block and adaptive pathways. Inspired by the idea of patch-

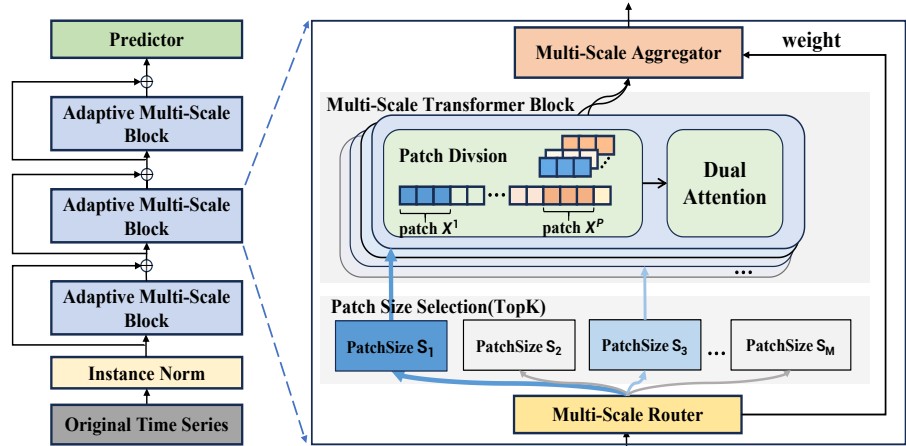

Figure 2: The architecture of Pathformer. The Multi-scale Transformer Block (MST Block) comprises patch division with multiple patch sizes and dual attention. The adaptive pathways select the patch sizes with the top $K$ weights generated by the router to capture multi-scale characteristics, and the selected patch sizes are represented in blue. Then, the aggregator applies weighted aggregation to the characteristics obtained from the MST Block.

ing in Transformers (Nie et al., 2023), *the multi-scale Transformer block* integrates multi-scale temporal resolutions and distances by introducing patch division with multiple patch sizes and dual attention on the divided patches, equipping the model with the capability to comprehensively model multi-scale characteristics. Based on various options of multi-scale modeling in the Transformer block, *adaptive pathways* utilize the multi-scale modeling capability and endow it with adaptive modeling characteristics. A multi-scale router selects specific sizes of patch division and the subsequent dual attention in the Transformer based on the input data, which controls the extraction of multi-scale features. The router works with an aggregator to combine these multi-scale characteristics through weighted aggregation. The layer-by-layer routing and aggregation form the adaptive pathways of multi-scale modeling throughout the Transformer blocks. In the following parts, we describe the multi-scale Transformer block and the adaptive pathways of the AMS Block in detail.

## 3.1 MULTI-SCALE TRANSFORMER BLOCK

**Multi-scale Division.** For the simplicity of notations, we use a univariate time series for description, and the method can be easily extended to multivariate cases by considering each variable independently. In the multi-scale Transformer block, We define a collection of $M$ patch size values as $\mathcal{S} = \{S_1, \ldots, S_M\}$, with each patch size $S$ corresponding to a patch division operation. For the input time series $\mathrm{X} \in \mathbb{R}^{H \times d}$, where $H$ denotes the length of the time series and $d$ denotes the dimension of features, each patch division operation with the patch size $S$ divides X into $P$ (with $P = H/S$) patches as $(\mathrm{X}^1, \mathrm{X}^2, \ldots, \mathrm{X}^P)$, where each patch $\mathrm{X}^i \in \mathbb{R}^{S \times d}$ contains $S$ time steps. Different patch sizes in the collection lead to various scales of divided patches and give various views of temporal resolutions for the input series. This multi-scale division works with the dual attention mechanism described below for multi-scale modeling.

**Dual Attention.** Based on the patch division of each scale, we propose dual attention to model temporal dependencies over the divided patches. To grasp temporal dependencies from different temporal distances, we utilize patch division as guidance for different temporal distances, and the dual attention mechanism consists of *intra-patch* attention within each divided patch and *inter-patch* attention across different patches, as shown in Figure 3(a).

Consider a set of patches $(\mathrm{X}^1, \mathrm{X}^2, \ldots, \mathrm{X}^P)$ divided with the patch size $S$, *intra-patch* attention establishes relationships between time steps within each patch. For the $i$-th patch $\mathrm{X}^i \in \mathbb{R}^{S \times d}$, we first embed the patch along the feature dimension $d$ to get $X_{\mathrm{intra}}^i \in \mathbb{R}^{S \times d_m}$, where $d_m$ represents the dimension of embedding. Then we perform trainable linear transformations on $\mathrm{X}_{\mathrm{intra}}^i$ to obtain the key and value in attention operations, denoted as $K_{\mathrm{intra}}^i, V_{\mathrm{intra}}^i \in \mathbb{R}^{S \times d_m}$. We employ a trainable query matrix $Q_{\mathrm{intra}}^i \in \mathbb{R}^{1 \times d_m}$ to merge the context of the patch and subsequently compute the

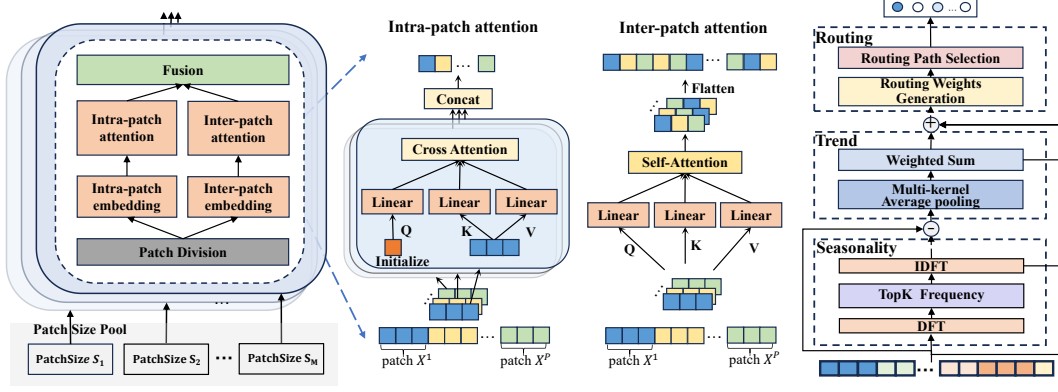

Figure 3: (a) The structure of the Multi-Scale Transformer Block, which mainly consists of Patch Division, Inter-patch attention, and Intra-patch attention. (b) The structure of the Multi-Scale Router.

cross-attention between $Q_{\text{intra}}^i, K_{\text{intra}}^i, V_{\text{intra}}^i$ to model local details within the $i$-th patch:

$$\text{Attn}_{\text{intra}}^i = \text{Softmax}(Q_{\text{intra}}^i(K_{\text{intra}}^i)^T/\sqrt{d_m})V_{\text{intra}}^i. \tag{1}$$

After intra-patch attention, each patch has transitioned from its original input length of $S$ to the length of $1$. The attention results from all the patches are concatenated to produce the output of intra-attention on the divided patches as $\text{Attn}_{\text{intra}} \in \mathbb{R}^{P \times d_m}$, which represents the local details from nearby time steps in the time series:

$$\text{Attn}_{\text{intra}} = \text{Concat}(\text{Attn}_{\text{intra}}^1, \dots, \text{Attn}_{\text{intra}}^P). \tag{2}$$

*Inter-patch* attention establishes relationships between patches to capture global correlations. For the patch-divided time series $\text{X} \in \mathbb{R}^{P \times S \times d}$, we first perform feature embedding along the feature dimension from $d$ to $d_m$ and then rearrange the data to combine the two dimensions of patch quantity $S$ and feature embedding $d_m$, resulting in $\text{X}_{\text{inter}} \in \mathbb{R}^{P \times d_m'}$, where $d_m' = S \cdot d_m$. After such embedding and rearranging process, the time steps within the same patch are combined, and thus we perform self-attention over $\text{X}_{\text{inter}}$ to model correlations between patches. Following the standard self-attention protocol, we obtain the query, key, and value through linear mapping on $\text{X}_{\text{inter}}$, denoted as $Q_{\text{inter}}, K_{\text{inter}}, V_{\text{inter}} \in \mathbb{R}^{P \times d_m'}$. Then, we compute the attention $\text{Attn}_{\text{inter}}$, which involves interaction between patches and represents the global correlations of the time series:

$$\text{Attn}_{\text{inter}} = \text{Softmax}(Q_{\text{inter}}(K_{\text{inter}})^T/\sqrt{d_m'})V_{\text{inter}}. \tag{3}$$

To fuse global correlations and local details captured by dual attention, we rearrange the outputs of intra-patch attention to $\text{Attn}_{\text{intra}} \in \mathbb{R}^{P \times S \times d_m}$, performing linear transformations on the patch size dimension from $1$ to $S$, to combine time steps in each patch, and then add it with inter-patch attention $\text{Attn}_{\text{inter}} \in \mathbb{R}^{P \times S \times d_m}$ to obtain the final output of dual attention $\text{Attn} \in \mathbb{R}^{P \times S \times d_m}$.

Overall, the multi-scale division provides different views of the time series with different patch sizes, and the changing patch sizes further influence the dual attention, which models temporal dependencies from different distances guided by the patch division. These two components work together to enable multiple scales of temporal modeling in the Transformer.

## 3.2 ADAPTIVE PATHWAYS

The design of the multi-scale Transformer block equips the model with the capability of multi-scale modeling. However, different series may prefer diverse scales, depending on their specific temporal characteristics and dynamics. Simply applying more scales may bring in redundant or useless signals, and manually tuning the optimal scales for a dataset or each time series is time-consuming or intractable. An ideal model needs to figure out such critical scales based on the input data for more effective modeling and better generalization of unseen data.

Pathways and Mixture of Experts are used to achieve adaptive modeling (Dean, 2021; Shazeer et al., 2016). Based on these concepts, we propose adaptive pathways based on multi-scale Transformer to model adaptive multi-scale, depicted in Figure 2. It contains two main components: the multi-scale router and the multi-scale aggregator. The *multi-scale router* selects specific sizes of patch division based on the input data, which activates specific parts in the Transformer and controls the extraction of multi-scale characteristics. The router works with the *multi-scale aggregator* to combine these characteristics through weighted aggregation, obtaining the output of the Transformer block.

**Multi-Scale Router.** The multi-scale router enables data-adaptive routing in the multi-scale Transformer, which selects the optimal sizes for patch division and thus controls the process of multi-scale modeling. Since the optimal or critical scales for each time series can be impacted by its complex inherent characteristics and dynamic patterns, like the periodicity and trend, we introduce a temporal decomposition module in the router that encompasses both *seasonality and trend decomposition* to extract periodicity and trend patterns, as illustrated in Figure 3(b).

*Seasonality decomposition* involves transforming the time series from the temporal domain into the frequency domain to extract the periodic patterns. We utilize the Discern Fourier Transform (DFT) (Cooley & Tukey, 1965), denoted as $\mathrm{DFT}(\cdot)$, to decompose the input X into Fourier basis and select the $K_f$ basis with the largest amplitudes to keep the sparsity of frequency domain. Then, we obtain the periodic patterns $\mathrm{X_{sea}}$ through an inverse DFT, denoted as $\mathrm{IDFT}(\cdot)$. The process is as follows:

$$\mathrm{X_{sea}} = \mathrm{IDFT}(\{f_1, \ldots, f_{K_f}\}, A, \Phi), \tag{4}$$

where $\Phi$ and $A$ represent the phase and amplitude of each frequency from $\mathrm{DFT(X)}$, $\{f_1, \ldots, f_{K_f}\}$ represents the frequencies with the top $K_f$ amplitudes. *Trend decomposition* uses different kernels of average pooling for moving averages to extract trend patterns based on the remaining part after the seasonality decomposition $\mathrm{X_{rem}} = \mathrm{X} - \mathrm{X_{sea}}$. For the results obtained from different kernels, a weighted operation is applied to obtain the representation of the trend component:

$$\mathrm{X_{trend}} = \mathrm{Softmax}(L(\mathrm{X_{rem}})) \cdot (\mathrm{Avgpool}(\mathrm{X_{rem}})_{\mathrm{kernel}_1}, \ldots, \mathrm{Avgpool}(\mathrm{X_{rem}})_{\mathrm{kernel}_N}), \tag{5}$$

where $\mathrm{Avgpool}(\cdot)_{\mathrm{kernel}_i}$ is the pooling function with the $i$-th kernel, $N$ corresponds to the number of kernels, $\mathrm{Softmax}(L(\cdot))$ controls the weights for the results from different kenerls. We add the seasonality pattern and trend pattern with the original input X, and then perform a linear mapping $\mathrm{Linear}(\cdot)$ to transform and merge them along the temporal dimension to get $\mathrm{X_{trans}} \in \mathbb{R}^d$.

Based on the results $\mathrm{X_{trans}}$ from temporal decomposition, the router employs a routing function to generate the pathway weights, which determines the patch sizes to choose for the current data. To avoid consistently selecting a few patch sizes, causing the corresponding scales to be repeatedly updated while neglecting other potentially useful scales in the multi-scale Transformer, we introduce noise terms to add randomness in the weight generation process. The whole process of generating pathway weights is as follows:

$$R(\mathrm{X_{trans}}) = \mathrm{Softmax}(\mathrm{X_{trans}}W_r + \epsilon \cdot \mathrm{Softplus}(\mathrm{X_{trans}}W_{\mathrm{noise}})), \epsilon \sim \mathcal{N}(0, 1), \tag{6}$$

where $R(\cdot)$ represents the whole routing function, $W_r$ and $W_{\mathrm{noise}} \in \mathbb{R}^{d \times M}$ are learnable parameters for weight generation, with $d$ denoting the feature dimension of $\mathrm{X_{trans}}$ and $M$ denoting the number of patch sizes. To introduce sparsity in the routing and encourage the selection of critical scales, we perform top$K$ selection on the pathway weights, keeping the top $K$ pathway weights and setting the rest weights as 0, and denote the final result as $\bar{R}(\mathrm{X_{trans}})$.

**Multi-Scale Aggregator.** Each dimension of the generated pathway weights $\bar{R}(\mathrm{X_{trans}}) \in \mathbb{R}^M$ correspond to a patch size in the multi-scale Transformer, with $\bar{R}(\mathrm{X_{trans}})_i > 0$ indicating performing this size $S_i$ of patch division and the dual attention and $\bar{R}(\mathrm{X_{trans}})_i = 0$ indicating ignoring this patch size for the current data. Let $\mathrm{X_{out}^i}$ denote the output of the multi-scale Transformer with the patch size $S_i$, due to the varying temporal dimensions produced by different patch sizes, the aggregator first perform a transformation function $T_i(\cdot)$ to align the temporal dimension from different scales. Then, the aggregator performs weighted aggregation for the multi-scale outputs based on the pathway weights to get the final output of this AMS block:

$$\mathrm{X_{out}} = \sum_{i=1}^{M} \mathcal{I}(\bar{R}(\mathrm{X_{trans}})_i > 0)R(\mathrm{X_{trans}})_i T_i(\mathrm{X_{out}^i}). \tag{7}$$

$\mathcal{I}(\bar{R}(\mathrm{X_{trans}})_i > 0)$ is the indicator function which outputs 1 when $\bar{R}(\mathrm{X_{trans}})_i > 0$, and otherwise outputs 0, indicating that only the top $K$ patch sizes and the corresponding outputs from the Transformer are considered or needed during aggregation.

## 4 EXPERIMENTS

### 4.1 TIME SERIES FORECASTING

**Datasets.** We conduct experiments on nine real-world datasets to assess the performance of Pathformer, encompassing a range of domains, including electricity transportation, weather forecasting, and cloud computing. These datasets include ETT (ETTh1, ETTh2, ETTm1, ETTm2), Weather, Electricity, Traffic, ILI, and Cloud Cluster (Cluster-A, Cluster-B, Cluster-C).

**Baselines and Metrics.** We choose some state-of-the-art models to serve as baselines, including PatchTST (Nie et al., 2023), NLinear (Zeng et al., 2023), Scaleformer (Shabani et al., 2023), TIDE (Das et al., 2023), FEDformer (Zhou et al., 2022), Pyraformer (Liu et al., 2022b), and Autoformer (Wu et al., 2021). To ensure fair comparisons, all models follow the same input length ($H = 36$ for the ILI dataset and $H = 96$ for others) and prediction length ($F \in \{24, 49, 96, 192\}$ for Cloud Cluster datasets, $F \in \{24, 36, 48, 60\}$ for ILI dataset and $F \in \{96, 192, 336, 720\}$ for others). We select two common metrics in time series forecasting: Mean Absolute Error (MAE) and Mean Squared Error (MSE).

**Implementation Details.** Pathformer utilizes the Adam optimizer (Kingma & Ba, 2015) with a learning rate set at $10^{-3}$. The default loss function employed is L1 Loss, and we implement early stopping within 10 epochs during the training process. All experiments are conducted using PyTorch and executed on an NVIDIA A800 80GB GPU. Pathformer is composed of 3 Adaptive Multi-Scale Blocks (AMS Blocks). Each AMS Block contains 4 different patch sizes. These patch sizes are selected from a pool of commonly used options, namely $\{2, 3, 6, 12, 16, 24, 32\}$.

**Main Results.** Table 1 shows the prediction results of multivariable time series forecasting, where Pathformer stands out with the best performance in 81 cases and the second-best in 5 cases out of the overall 88 cases. Compared with the second-best baseline, PatchTST, Pathformer demonstrates a significant improvement, with an impressive 8.1% reduction in MSE and a 6.4% reduction in MAE. Compared with the strong linear models NLinear, Pathformer also outperforms them comprehensively, especially on large datasets such as Electricity and Traffic. This demonstrates the potential of Transformer architecture for time series forecasting. Compared with the multi-scale models Pyraformer and Scaleformer, Pathformer exhibits good performance improvements, with a substantial 36.4% reduction in MSE and a 19.1% reduction in MAE. This illustrates that the proposed comprehensive modeling from both temporal resolution and temporal distance with adaptive pathways is more effective for multi-scale modeling.

### 4.2 TRANSFER LEARNING

**Experimental Setting.** To assess the transferability of Pathformer, we benchmark it against three baselines: PatchTST, FEDformer, and Autoformer, devising two distinct transfer experiments. In the context of evaluating transferability across different datasets, models initially undergo pre-training on the ETTh1 and ETTm1. Subsequently, we fine-tune them using the ETTh2 and ETTm2. For assessing transferability towards future data, models are pre-trained on the first 70% of the training data sourced from three clusters: Cluster-A, Cluster-B, and Cluster-C. This pre-training is followed by fine-tuning the remaining 30% of the training data specific to each cluster. In terms of methodology for baselines, we explore two approaches: direct prediction (zero-shot) and full-tuning. Deviating from these approaches, Pathformer integrates a part-tuning strategy. In this approach, specific parameters, like those of the router network, undergo fine-tuning, resulting in a significant reduction in computational resource demands.

**Transfer Learning Results.** Table 2 presents the outcomes of our transfer learning evaluation. Across both direct prediction and full-tuning methods, Pathformer surpasses the baseline models, highlighting its enhanced generalization and transferability. One of the key strengths of Pathformer lies in its adaptive capacity to select varying scales for different temporal dynamics. This adaptability allows it to effectively capture complex temporal patterns present in diverse datasets, consequently demonstrating superior generalization and transferability. Part-tuning is a lightweight fine-tuning method that demands fewer computational resources and reduces training time on average by $52\%$, while still achieving prediction accuracy nearly comparable to Pathformer full-tuning. Moreover, it outperforms the full-tuning of other baseline models on the majority of datasets. This demonstrates that Pathformer can provide effective lightweight transfer learning for time series forecasting.

Table 1: Multivariate time series forecasting results. The input length $H = 96$ ($H = 36$ for ILI). The best results are highlighted in bold, and the second-best results are underlined.

| Method | | Pathformer | | PatchTST | | NLinear | | Scaleformer | | TiDE | | FEDformer | | Pyraformer | | Autoformer | |
|---|---|---|---|---|---|---|---|---|---|---|---|---|---|---|---|---|---|
| Metric | | MSE | MAE | MSE | MAE | MSE | MAE | MSE | MAE | MSE | MAE | MSE | MAE | MSE | MAE | MSE | MAE |
| ETTh1 | 96 | 0.382 | **0.400** | 0.394 | 0.408 | 0.386 | **0.392** | 0.396 | 0.440 | 0.427 | 0.450 | **0.376** | 0.419 | 0.664 | 0.612 | 0.449 | 0.459 |
| | 192 | **0.440** | **0.427** | 0.446 | 0.438 | 0.440 | 0.430 | 0.434 | 0.460 | 0.472 | 0.486 | **0.420** | 0.448 | 0.790 | 0.681 | 0.500 | 0.482 |
| | 336 | **0.454** | **0.432** | 0.485 | 0.455 | 0.480 | 0.443 | 0.462 | 0.476 | 0.527 | 0.527 | 0.459 | 0.465 | 0.891 | 0.738 | 0.521 | 0.496 |
| | 720 | **0.479** | **0.461** | 0.495 | 0.474 | 0.486 | 0.472 | 0.494 | 0.500 | 0.644 | 0.605 | 0.506 | 0.507 | 0.963 | 0.782 | 0.514 | 0.512 |
| ETTh2 | 96 | **0.279** | **0.331** | 0.294 | 0.343 | 0.290 | 0.339 | 0.364 | 0.407 | 0.304 | 0.359 | 0.346 | 0.388 | 0.645 | 0.597 | 0.358 | 0.397 |
| | 192 | **0.349** | **0.380** | 0.378 | 0.394 | 0.379 | 0.395 | 0.466 | 0.458 | 0.394 | 0.422 | 0.429 | 0.439 | 0.788 | 0.683 | 0.456 | 0.452 |
| | 336 | **0.348** | **0.382** | 0.382 | 0.410 | 0.421 | 0.431 | 0.479 | 0.476 | 0.385 | 0.421 | 0.496 | 0.487 | 0.907 | 0.747 | 0.482 | 0.486 |
| | 720 | **0.398** | **0.424** | 0.412 | 0.433 | 0.436 | 0.453 | 0.487 | 0.492 | 0.463 | 0.475 | 0.463 | 0.474 | 0.963 | 0.783 | 0.515 | 0.511 |
| ETTm1 | 96 | **0.316** | **0.346** | 0.324 | 0.361 | 0.339 | 0.369 | 0.355 | 0.398 | 0.356 | 0.381 | 0.379 | 0.419 | 0.543 | 0.510 | 0.505 | 0.475 |
| | 192 | 0.366 | **0.370** | **0.362** | 0.383 | 0.379 | 0.386 | 0.428 | 0.455 | 0.391 | 0.399 | 0.426 | 0.441 | 0.557 | 0.537 | 0.553 | 0.496 |
| | 336 | **0.386** | **0.394** | 0.390 | 0.402 | 0.411 | 0.407 | 0.524 | 0.487 | 0.424 | 0.423 | 0.445 | 0.459 | 0.754 | 0.655 | 0.621 | 0.537 |
| | 720 | **0.460** | **0.432** | 0.461 | 0.438 | 0.478 | 0.442 | 0.558 | 0.517 | 0.480 | 0.456 | 0.543 | 0.490 | 0.908 | 0.724 | 0.671 | 0.561 |
| ETTm2 | 96 | **0.170** | **0.248** | 0.177 | 0.260 | 0.177 | 0.257 | 0.182 | 0.275 | 0.182 | 0.264 | 0.203 | 0.287 | 0.435 | 0.507 | 0.255 | 0.339 |
| | 192 | **0.238** | **0.295** | 0.248 | 0.306 | 0.241 | 0.297 | 0.251 | 0.318 | 0.256 | 0.323 | 0.269 | 0.328 | 0.730 | 0.673 | 0.281 | 0.340 |
| | 336 | **0.293** | **0.331** | 0.304 | 0.342 | 0.302 | 0.337 | 0.340 | 0.375 | 0.313 | 0.354 | 0.325 | 0.366 | 1.201 | 0.845 | 0.339 | 0.372 |
| | 720 | **0.390** | **0.389** | 0.403 | 0.397 | 0.405 | 0.396 | 0.435 | 0.433 | 0.419 | 0.410 | 0.421 | 0.415 | 3.625 | 1.451 | 0.433 | 0.432 |
| Weather | 96 | **0.156** | **0.192** | 0.177 | 0.218 | 0.168 | 0.208 | 0.288 | 0.365 | 0.202 | 0.261 | 0.238 | 0.314 | 0.896 | 0.556 | 0.249 | 0.329 |
| | 192 | **0.206** | **0.240** | 0.224 | 0.258 | 0.217 | 0.255 | 0.368 | 0.425 | 0.242 | 0.298 | 0.275 | 0.329 | 0.622 | 0.624 | 0.325 | 0.370 |
| | 336 | **0.254** | **0.282** | 0.277 | 0.297 | 0.267 | 0.292 | 0.447 | 0.469 | 0.287 | 0.335 | 0.339 | 0.377 | 0.739 | 0.753 | 0.351 | 0.391 |
| | 720 | **0.340** | **0.336** | 0.350 | 0.345 | 0.351 | 0.346 | 0.640 | 0.574 | 0.351 | 0.386 | 0.389 | 0.409 | 1.004 | 0.934 | 0.415 | 0.426 |
| Electricity | 96 | **0.145** | **0.236** | 0.180 | 0.264 | 0.185 | 0.266 | 0.182 | 0.297 | 0.194 | 0.277 | 0.186 | 0.302 | 0.386 | 0.449 | 0.196 | 0.313 |
| | 192 | **0.167** | **0.256** | 0.188 | 0.275 | 0.189 | 0.276 | 0.188 | 0.300 | 0.193 | 0.280 | 0.197 | 0.311 | 0.386 | 0.443 | 0.211 | 0.324 |
| | 336 | **0.186** | **0.275** | 0.206 | 0.291 | 0.204 | 0.289 | 0.210 | 0.324 | 0.206 | 0.296 | 0.213 | 0.328 | 0.378 | 0.443 | 0.214 | 0.327 |
| | 720 | **0.231** | **0.309** | 0.247 | 0.328 | 0.245 | 0.319 | 0.232 | 0.339 | 0.242 | 0.328 | 0.233 | 0.344 | 0.376 | 0.445 | 0.236 | 0.342 |
| ILI | 24 | **1.587** | **0.758** | 1.724 | 0.843 | 2.725 | 1.069 | 0.232 | 0.339 | 2.154 | 0.992 | 2.624 | 1.095 | 1.420 | 2.012 | 2.906 | 1.182 |
| | 36 | **1.429** | **0.711** | 1.536 | 0.752 | 2.530 | 1.032 | 2.745 | 1.075 | 2.436 | 1.042 | 2.516 | 1.021 | 7.394 | 2.031 | 2.585 | 1.038 |
| | 48 | **1.505** | **0.742** | 1.821 | 0.832 | 2.510 | 1.031 | 2.748 | 1.072 | 2.532 | 1.051 | 2.505 | 1.041 | 7.551 | 2.057 | 3.024 | 1.145 |
| | 60 | **1.731** | **0.799** | 1.923 | 0.842 | 2.492 | 1.026 | 2.793 | 1.059 | 2.748 | 1.142 | 2.742 | 1.122 | 7.662 | 2.100 | 2.761 | 1.114 |
| Traffic | 96 | **0.479** | **0.283** | 0.492 | 0.324 | 0.645 | 0.388 | 2.678 | 1.071 | 0.568 | 0.352 | 0.576 | 0.359 | 2.085 | 0.468 | 0.597 | 0.371 |
| | 192 | **0.484** | **0.292** | 0.487 | 0.303 | 0.599 | 0.365 | 0.564 | 0.351 | 0.612 | 0.371 | 0.610 | 0.380 | 0.867 | 0.467 | 0.607 | 0.382 |
| | 336 | **0.503** | **0.299** | 0.505 | 0.317 | 0.606 | 0.367 | 0.570 | 0.349 | 0.605 | 0.374 | 0.608 | 0.375 | 0.869 | 0.469 | 0.623 | 0.387 |
| | 720 | **0.537** | **0.322** | 0.542 | 0.337 | 0.645 | 0.388 | 0.576 | 0.349 | 0.647 | 0.410 | 0.621 | 0.375 | 0.881 | 0.473 | 0.639 | 0.395 |
| Cluster-A | 24 | **0.100** | **0.205** | 0.126 | 0.234 | 0.134 | 0.235 | 0.128 | 0.247 | 0.128 | 0.244 | 0.131 | 0.260 | 0.131 | 0.268 | 0.372 | 0.461 |
| | 48 | **0.160** | **0.264** | 0.208 | 0.302 | 0.214 | 0.310 | 0.182 | 0.319 | 0.192 | 0.299 | 0.175 | 0.307 | 0.170 | 0.311 | 0.390 | 0.471 |
| | 96 | **0.227** | **0.321** | 0.313 | 0.372 | 0.335 | 0.410 | 0.274 | 0.328 | 0.247 | 0.338 | 0.293 | 0.349 | 0.243 | 0.375 | 0.466 | 0.514 |
| | 192 | **0.349** | **0.400** | 0.452 | 0.453 | 0.442 | 0.452 | 0.372 | 0.451 | 0.356 | 0.422 | 0.350 | 0.439 | 0.378 | 0.437 | 0.585 | 0.584 |
| Cluster-B | 24 | **0.121** | **0.224** | 0.126 | 0.237 | 0.130 | 0.241 | 0.125 | 0.241 | 0.128 | 0.240 | 0.128 | 0.243 | 0.129 | 0.263 | 0.242 | 0.369 |
| | 48 | 0.172 | **0.270** | 0.183 | 0.290 | 0.173 | 0.285 | 0.164 | 0.280 | 0.165 | 0.288 | **0.156** | 0.287 | 0.168 | 0.296 | 0.299 | 0.425 |
| | 96 | **0.242** | **0.322** | 0.272 | 0.352 | 0.281 | 0.365 | 0.252 | 0.342 | 0.244 | 0.334 | 0.277 | 0.389 | 0.315 | 0.436 | 0.366 | 0.471 |
| | 192 | 0.437 | **0.427** | 0.476 | 0.461 | 0.479 | 0.456 | 0.438 | 0.447 | 0.452 | 0.467 | **0.414** | 0.478 | 0.389 | 0.485 | 0.597 | 0.563 |
| Cluster-C | 24 | **0.064** | **0.169** | 0.075 | 0.188 | 0.100 | 0.205 | 0.074 | 0.204 | 0.082 | 0.199 | 0.076 | 0.212 | 0.107 | 0.247 | 0.189 | 0.341 |
| | 48 | **0.102** | **0.218** | 0.118 | 0.241 | 0.163 | 0.286 | 0.110 | 0.242 | 0.121 | 0.266 | 0.108 | 0.246 | 0.142 | 0.284 | 0.210 | 0.363 |
| | 96 | **0.162** | **0.276** | 0.188 | 0.305 | 0.245 | 0.318 | 0.177 | 0.321 | 0.201 | 0.305 | 0.171 | 0.323 | 0.181 | 0.328 | 0.289 | 0.421 |
| | 192 | **0.304** | **0.369** | 0.354 | 0.413 | 0.375 | 0.457 | 0.326 | 0.428 | 0.341 | 0.424 | 0.338 | 0.453 | 0.332 | 0.396 | 0.419 | 0.511 |

Table 2: Transfer Learning results. The best results are in bold, and the second results are underlined.

| Models | | Pathformer | | | | | | PatchTST | | | | FEDformer | | | | Autoformer | | | |
|---|---|---|---|---|---|---|---|---|---|---|---|---|---|---|---|---|---|---|---|
| | | Predict | | Part-tuning | | Full-tuning | | Predict | | Full-tuning | | Predict | | Full-tuning | | Predict | | Full-tuning | |
| Metric | | MSE | MAE | MSE | MAE | MSE | MAE | MSE | MAE | MSE | MAE | MSE | MAE | MSE | MAE | MSE | MAE | MSE | MAE |
| ETTh2 | 96 | 0.340 | 0.369 | 0.287 | 0.333 | **0.276** | **0.328** | 0.346 | 0.369 | 0.287 | 0.337 | 0.420 | 0.449 | 0.326 | 0.337 | 0.397 | 0.439 | 0.342 | 0.386 |
| | 192 | 0.411 | 0.406 | 0.358 | 0.382 | **0.350** | **0.376** | 0.422 | 0.420 | 0.366 | 0.385 | 0.475 | 0.475 | 0.409 | 0.430 | 0.543 | 0.511 | 0.415 | 0.428 |
| | 336 | 0.384 | 0.401 | 0.342 | 0.384 | **0.337** | **0.374** | 0.408 | 0.419 | 0.377 | 0.405 | 0.416 | 0.446 | 0.378 | 0.416 | 0.521 | 0.515 | 0.415 | 0.442 |
| | 720 | 0.450 | 0.448 | 0.416 | 0.437 | **0.401** | **0.426** | 0.479 | 0.467 | 0.410 | 0.432 | 0.529 | 0.517 | 0.46 | 0.487 | 0.694 | 0.602 | 0.452 | 0.469 |
| ETTm2 | 96 | 0.220 | 0.294 | 0.181 | 0.260 | **0.172** | **0.251** | 0.189 | 0.284 | 0.177 | 0.261 | 0.256 | 0.378 | 0.201 | 0.285 | 0.331 | 0.406 | 0.212 | 0.293 |
| | 192 | 0.258 | 0.306 | 0.240 | 0.299 | **0.237** | **0.294** | 0.263 | 0.322 | 0.243 | 0.294 | 0.427 | 0.441 | 0.266 | 0.324 | 0.435 | 0.461 | 0.275 | 0.331 |
| | 336 | 0.325 | 0.350 | 0.305 | 0.339 | **0.302** | **0.334** | 0.332 | 0.365 | 0.305 | 0.339 | 0.429 | 0.448 | 0.335 | 0.369 | 0.506 | 0.501 | 0.333 | 0.370 |
| | 720 | 0.422 | 0.408 | 0.406 | 0.398 | **0.391** | **0.392** | 0.429 | 0.419 | 0.405 | 0.395 | 0.530 | 0.503 | 0.423 | 0.417 | 0.680 | 0.573 | 0.444 | 0.433 |
| Cluster-A | 24 | 0.121 | 0.223 | 0.100 | 0.205 | **0.097** | **0.202** | 0.143 | 0.250 | 0.115 | 0.221 | 0.200 | 0.326 | 0.171 | 0.298 | 0.382 | 0.471 | 0.349 | 0.445 |
| | 48 | 0.186 | 0.281 | 0.159 | 0.261 | **0.144** | **0.254** | 0.231 | 0.322 | 0.192 | 0.289 | 0.240 | 0.360 | 0.219 | 0.342 | 0.372 | 0.463 | 0.362 | 0.450 |
| | 96 | 0.249 | 0.334 | 0.215 | 0.313 | **0.193** | **0.302** | 0.350 | 0.396 | 0.290 | 0.349 | 0.326 | 0.418 | 0.299 | 0.392 | 0.395 | 0.490 | 0.375 | 0.432 |
| | 192 | 0.372 | 0.416 | 0.312 | 0.381 | **0.292** | **0.371** | 0.524 | 0.491 | 0.406 | 0.433 | 0.381 | 0.463 | 0.338 | 0.432 | 0.948 | 0.761 | 0.592 | 0.602 |
| Cluster-B | 24 | 0.140 | 0.243 | 0.120 | 0.226 | **0.117** | **0.221** | 0.145 | 0.248 | 0.124 | 0.231 | 0.167 | 0.283 | 0.147 | 0.271 | 0.226 | 0.342 | 0.192 | 0.318 |
| | 48 | 0.202 | 0.298 | 0.174 | 0.275 | 0.170 | **0.270** | 0.207 | 0.306 | 0.178 | 0.282 | 0.225 | 0.310 | **0.162** | 0.283 | 0.247 | 0.361 | 0.234 | 0.354 |
| | 96 | 0.296 | 0.357 | 0.253 | 0.327 | **0.244** | **0.321** | 0.298 | 0.365 | 0.264 | 0.242 | 0.347 | 0.427 | 0.318 | 0.408 | 0.307 | 0.430 | 0.280 | 0.399 |
| | 192 | 0.464 | 0.468 | 0.441 | 0.425 | **0.425** | **0.420** | 0.529 | 0.495 | 0.471 | 0.463 | 0.528 | 0.497 | 0.434 | 0.478 | 0.618 | 0.614 | 0.584 | 0.578 |
| Cluster-C | 24 | 0.069 | 0.173 | 0.064 | 0.166 | **0.062** | **0.165** | 0.074 | 0.184 | 0.072 | 0.182 | 0.109 | 0.243 | 0.097 | 0.229 | 0.212 | 0.344 | 0.194 | 0.332 |
| | 48 | 0.144 | 0.254 | 0.104 | 0.219 | **0.101** | **0.215** | 0.138 | 0.246 | 0.115 | 0.233 | 0.150 | 0.285 | 0.118 | 0.260 | 0.228 | 0.366 | 0.214 | 0.362 |
| | 96 | 0.174 | 0.284 | 0.166 | 0.275 | **0.162** | **0.272** | 0.194 | 0.303 | 0.182 | 0.298 | 0.228 | 0.342 | 0.190 | 0.325 | 0.281 | 0.436 | 0.263 | 0.405 |
| | 192 | 0.327 | 0.386 | 0.316 | 0.374 | **0.301** | **0.365** | 0.376 | 0.413 | 0.349 | 0.407 | 0.344 | 0.444 | 0.332 | 0.441 | 0.508 | 0.537 | 0.417 | 0.507 |

## 4.3 ABLATION STUDIES

To ascertain the impact of different modules within Pathformer, we perform ablation studies focusing on inter-patch attention, intra-patch attention, time series decomposition, and Pathways. The W/O Pathways configuration entails using all patch sizes from the patch size pool for every dataset, eliminating adaptive selection. Table 3 illustrates the unique impact of each module. The influence of Pathways is significant; omitting them results in a marked decrease in prediction accuracy. This emphasizes the criticality of optimizing the mix of patch sizes to extract multi-scale characteristics, thus markedly improving the model's prediction accuracy. Regarding efficiency, intra-patch attention is notably adept at discerning local patterns, contrasting with inter-patch attention which primarily captures wider global patterns. The time series decomposition module decomposes trend

Table 3: Ablation study. W/O Inter, W/O Intra, W/O Decompose represent removing the inter-patch attention, intra-patch attention, and time series decomposition, respectively.

| Models | | W/O Inter | | W/O Intra | | W/O Decompose | | W/O Pathways | | Pathformer | |
|---|---|---|---|---|---|---|---|---|---|---|---|---|
| Metric | | MSE | MAE | MSE | MAE | MSE | MAE | MSE | MAE | MSE | MAE |
| Weather | 96 | 0.162 | 0.196 | 0.170 | 0.203 | 0.162 | 0.198 | 0.168 | 0.204 | **0.156** | **0.192** |
| | 192 | 0.219 | 0.248 | 0.220 | 0.249 | 0.212 | 0.244 | 0.219 | 0.250 | **0.206** | **0.240** |
| | 336 | 0.262 | 0.290 | 0.272 | 0.292 | 0.256 | 0.285 | 0.269 | 0.290 | **0.254** | **0.282** |
| | 720 | 0.350 | 0.349 | 0.358 | 0.357 | 0.344 | 0.340 | 0.349 | 0.348 | **0.340** | **0.336** |
| Electricity | 96 | 0.166 | 0.259 | 0.182 | 0.264 | 0.152 | 0.244 | 0.168 | 0.256 | **0.145** | **0.236** |
| | 192 | 0.185 | 0.270 | 0.193 | 0.275 | 0.176 | 0.264 | 0.185 | 0.272 | **0.167** | **0.256** |
| | 336 | 0.216 | 0.301 | 0.214 | 0.297 | 0.195 | 0.281 | 0.210 | 0.296 | **0.186** | **0.275** |
| | 720 | 0.239 | 0.322 | 0.253 | 0.327 | 0.235 | 0.316 | 0.254 | 0.332 | **0.231** | **0.309** |

Table 4: Parameter sensitivity study. The prediction accuracy varies with $K$.

| | | $K=1$ | | $K=2$ | | $K=3$ | | $K=4$ | |
|---|---|---|---|---|---|---|---|---|---|
| Metric | | MSE | MAE | MSE | MAE | MSE | MAE | MSE | MAE |
| ETTh2 | 96 | 0.283 | 0.333 | **0.279** | **0.331** | 0.286 | 0.337 | 0.282 | 0.333 |
| | 192 | 0.357 | 0.380 | **0.349** | **0.380** | 0.354 | 0.383 | 0.359 | 0.384 |
| | 336 | 0.342 | 0.379 | 0.348 | 0.382 | **0.338** | **0.377** | 0.347 | 0.380 |
| | 720 | 0.411 | 0.430 | **0.398** | **0.424** | 0.406 | 0.428 | 0.407 | 0.432 |
| Electricity | 96 | 0.162 | 0.247 | **0.145** | **0.236** | 0.147 | 0.238 | 0.152 | 0.244 |
| | 192 | 0.175 | 0.260 | **0.167** | **0.256** | 0.176 | 0.265 | 0.178 | 0.266 |
| | 336 | 0.192 | 0.278 | 0.186 | 0.275 | **0.181** | **0.274** | 0.190 | 0.277 |
| | 720 | 0.234 | 0.311 | 0.231 | 0.309 | **0.230** | **0.308** | 0.235 | 0.313 |

and periodic patterns to improve the ability to capture the temporal dynamics of its input, assisting in the identification of appropriate patch sizes for combination.

**Varying the Number of Adaptively Selected Patch Sizes.** Pathformer adaptively selects the top $K$ patch sizes for combination, adjusting to different time series samples. We evaluate the influence of different $K$ values on prediction accuracy in Table 4. Our findings show that $K = 2$ and $K = 3$ yield better results than $K = 1$ and $K = 4$, highlighting the advantage of adaptively modeling critical multi-scale characteristics for improved accuracy. Additionally, distinct time series samples benefit from feature extraction using varied patch sizes, but not all patch sizes are equally effective.

**Visualization of Pathways Weights.** We show three samples and depict their average Pathways weights for each patch size in Figure 4. Our observations reveal that the samples possess unique Pathways weight distributions. Both Samples 1 and 2, which demonstrate longer seasonality and similar trend patterns, show similar visualized Pathways weights. This manifests in the higher weights they attribute to the larger patch sizes. On the other hand, Sample 3, which is characterized by its shorter seasonality pattern, aligns with higher weights for the smaller patch sizes. These observations underscore Pathformer's adaptability, emphasizing its ability to discern and apply the optimal patch size combinations for the diverse seasonality and trend patterns across samples.

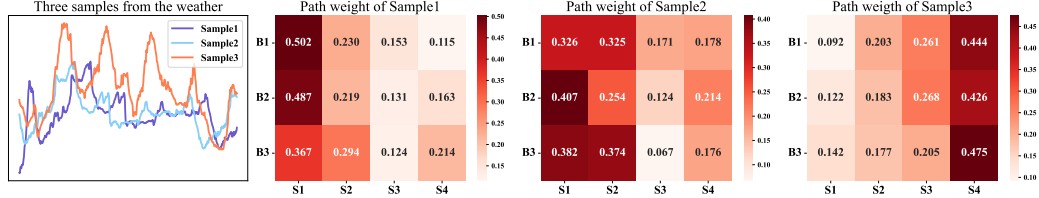

Figure 4: The average pathways weights of different patch sizes for the Weather. $B_1$, $B_2$, and $B_3$ denote distinct AMS (Adaptive Multi-Scale) blocks, while $S_1$, $S_2$, $S_3$, and $S_4$ represent varying patch sizes within each AMS block, with patch size decreasing sequentially.

## 5 CONCLUSION

In this paper, we propose Pathformer, a Multi-Scale Transformer with Adaptive Pathways for time series forecasting. It integrates multi-scale temporal resolutions and temporal distances by introducing patch division with multiple patch sizes and dual attention on the divided patches, enabling the comprehensive modeling of multi-scale characteristics. Furthermore, adaptive pathways dynamically select and aggregate scale-specific characteristics based on the different temporal dynamics. These innovative mechanisms collectively empower Pathformer to achieve outstanding prediction performance and demonstrate strong generalization capability on several forecasting tasks.

ACKNOWLEDGMENTS

This work was supported by National Natural Science Foundation of China (62372179) and Alibaba Innovative Research Program.

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

# A APPENDIX

## A.1 EXPERIMENTAL DETAILS

### A.1.1 DATASETS

The Special details about experiment datasets are as follows: ETT [1] datasets consist of 7 variables, originating from two different electric transformers. It covers the period from January 2016 to January 2018. Each electric transformer has data recorded at 15-minute and 1-hour granularities, labeled as ETTh1, ETTh2, ETTm1, and ETTm2. Weather [2] dataset comprises 21 meteorological indicators in Germany, collected every 10 minutes. Electricity [3] dataset contains the power consumption of 321 users, recorded every hour, spanning from July 2016 to July 2019. ILI [4] collects weekly data on patients with influenza-like illness from the Centers for Disease Control and Prevention of the United States spanning the years 2002 to 2021. Traffic [5] comprises hourly data sourced from the California Department of Transportation. This dataset delineates road occupancy rates measured by various sensors on the freeways of the San Francisco Bay area. Cloud cluster datasets are private business data, documenting customer resource demands at 1-minute intervals for three clusters: cluster-A, cluster-B, cluster-C, where A,B,C represent different cities, covering the period from February 2023 to April 2023. For dataset preparation, we follow the established practice from previous studies (Zhou et al., 2021; Wu et al., 2021). Detailed statistics are shown in Table 5.

Table 5: The statistics of datasets

| Datasets | ETTh1&ETTh2 | ETTm1&ETTm2 | Weather | Electricity | ILI | Traffic | Cluster |
|---|---|---|---|---|---|---|---|
| **Variables** | 7 | 7 | 21 | 321 | 7 | 862 | 6 |
| **Timestamps** | 17420 | 69680 | 52696 | 26304 | 966 | 17544 | 256322 |
| **Split Ratio** | 6:2:2 | 6:2:2 | 7:1:2 | 7:1:2 | 7:1:2 | 7:1:2 | 7:1:2 |

### A.1.2 BASELINES

In the realm of time series forecasting, numerous models have surfaced in recent years. We choose models with superior predictive performance from 2021 to 2023 as baselines, including the 2021 state-of-the-art (SOTA) Autoformer, the 2022 SOTA FEDformer, and the 2023 SOTA PatchTST and NLinear, among others. The specific code repositories for each of these models are as follows:

- PatchTST: https://github.com/yuqinie98/PatchTST
- NLinear: https://github.com/cure-lab/LTSF-Linear
- FEDformer: https://github.com/MAZiqing/FEDformer
- Scaleformer: https://github.com/borealisai/scaleformer
- TiDE: https://github.com/google-research/google-research/tree/master/tide
- Pyraformer: https://github.com/ant-research/Pyraformer
- Autoformer: https://github.com/thuml/Autoformer

## A.2 UNIVARIATE TIME SERIES FORECASTING

We conducted univariate time series forecasting experiments on the ETT and Cloud cluster datasets. As shown in Table 6, Pathformer stands out with the best performance in 50 cases and as the second-best in 5 out of 56 instances. Pathformer has outperformed the second-best baseline PatchTST, especially on the Cloud cluster datasets. Our model Pathformer demonstrates excellent predictive performance in both multivariate and univariate time series forecasting.

---

[1]https://github.com/zhouhaoyi/ETDataset
[2]https://www.bgc-jena.mpg.de/wetter/
[3]https://archive.ics.uci.edu/ml/datasets/ElectricityLoadDiagrams20112014
[4]https://gis.cdc.gov/grasp/fluview/fluportaldashboard.html
[5]https://pems.dot.ca.gov/

Table 6: Univariate time series forecasting results. The input length $H = 96$, and the prediction length $F \in \{96, 192, 336, 720\}$(for cloud clusters datasets $F \in \{24, 48, 96, 192\}$). The best results are highlighted in bold.

| Models | | Pathformer | | PatchTST | | FEDformer | | Autoformer | |
|---|---|---|---|---|---|---|---|---|---|
| Metric | | MSE | MAE | MSE | MAE | MSE | MAE | MSE | MAE |
| ETTh1 | 96 | **0.057** | 0.180 | 0.057 | **0.179** | 0.079 | 0.215 | 0.071 | 0.206 |
| | 192 | **0.075** | **0.208** | 0.076 | 0.209 | 0.104 | 0.245 | 0.114 | 0.262 |
| | 336 | **0.076** | **0.216** | 0.093 | 0.240 | 0.119 | 0.270 | 0.107 | 0.258 |
| | 720 | **0.090** | **0.238** | 0.097 | 0.245 | 0.142 | 0.299 | 0.126 | 0.283 |
| ETTh2 | 96 | 0.128 | 0.274 | **0.127** | 0.273 | 0.128 | **0.271** | 0.153 | 0.306 |
| | 192 | **0.177** | 0.330 | 0.178 | **0.328** | 0.185 | 0.330 | 0.204 | 0.351 |
| | 336 | **0.180** | **0.340** | 0.221 | 0.374 | 0.231 | 0.378 | 0.246 | 0.389 |
| | 720 | **0.213** | **0.371** | 0.250 | 0.403 | 0.278 | 0.420 | 0.268 | 0.409 |
| ETTm1 | 96 | **0.029** | **0.126** | 0.030 | 0.127 | 0.033 | 0.140 | 0.056 | 0.183 |
| | 192 | **0.042** | **0.160** | 0.043 | 0.165 | 0.058 | 0.186 | 0.081 | 0.216 |
| | 336 | **0.058** | **0.185** | 0.059 | 0.185 | 0.084 | 0.231 | 0.076 | 0.218 |
| | 720 | **0.079** | **0.217** | 0.081 | 0.218 | 0.102 | 0.250 | 0.110 | 0.267 |
| ETTm2 | 96 | **0.062** | **0.179** | 0.064 | 0.181 | 0.072 | 0.206 | 0.065 | 0.189 |
| | 192 | **0.096** | **0.230** | 0.097 | 0.231 | 0.102 | 0.245 | 0.118 | 0.256 |
| | 336 | **0.128** | **0.268** | 0.129 | 0.270 | 0.130 | 0.279 | 0.154 | 0.305 |
| | 720 | **0.179** | **0.326** | 0.181 | 0.330 | 0.178 | 0.325 | 0.182 | 0.335 |
| Cluster-A | 24 | **0.137** | **0.218** | 0.174 | 0.256 | 0.203 | 0.303 | 0.455 | 0.483 |
| | 48 | **0.218** | **0.280** | 0.299 | 0.343 | 0.308 | 0.364 | 0.508 | 0.504 |
| | 96 | **0.298** | **0.337** | 0.434 | 0.409 | 0.361 | 0.403 | 0.563 | 0.524 |
| | 192 | **0.390** | **0.401** | 0.589 | 0.480 | 0.409 | 0.447 | 0.669 | 0.583 |
| Cluster-B | 24 | **0.100** | **0.206** | 0.107 | 0.218 | 0.130 | 0.253 | 0.197 | 0.339 |
| | 48 | **0.146** | **0.251** | 0.158 | 0.265 | 0.149 | 0.272 | 0.247 | 0.390 |
| | 96 | 0.219 | **0.301** | 0.234 | 0.327 | 0.230 | 0.342 | 0.313 | 0.429 |
| | 192 | 0.454 | **0.404** | 0.461 | 0.444 | **0.415** | 0.412 | 0.512 | 0.544 |
| Cluster-C | 24 | **0.080** | **0.191** | 0.092 | 0.210 | 0.120 | 0.258 | 0.206 | 0.354 |
| | 48 | **0.117** | **0.232** | 0.138 | 0.261 | 0.151 | 0.302 | 0.229 | 0.365 |
| | 96 | **0.176** | **0.286** | 0.222 | 0.330 | 0.198 | 0.342 | 0.293 | 0.420 |
| | 192 | **0.345** | **0.390** | 0.404 | 0.443 | 0.361 | 0.444 | 0.441 | 0.524 |

## A.3 VARYING THE INPUT LENGTH WITH TRANSFORMER MODELS

In time series forecasting tasks, the size of the input length determines how much historical information the model receives. We select models with better predictive performance from the main experiments as baselines. We configure different input lengths to evaluate the effectiveness of Pathformer and visualize the prediction results for input lengths of 48,192. From Figure 5, Pathformer consistently outperforms the baselines on the ETTh1, ETTh2, Weather, and Electricity. As depicted in Table 7 and Table 8, for $H = 48, 192$, Pathformer stands out with the best performance in 46, 44 cases out of 48, respectively. Based on the results above, it is evident that Pathformer outperforms the baselines across different input lengths. As the input length increases, the prediction metrics of Pathformer continue to decrease, indicating that it is capable of modeling longer sequences.

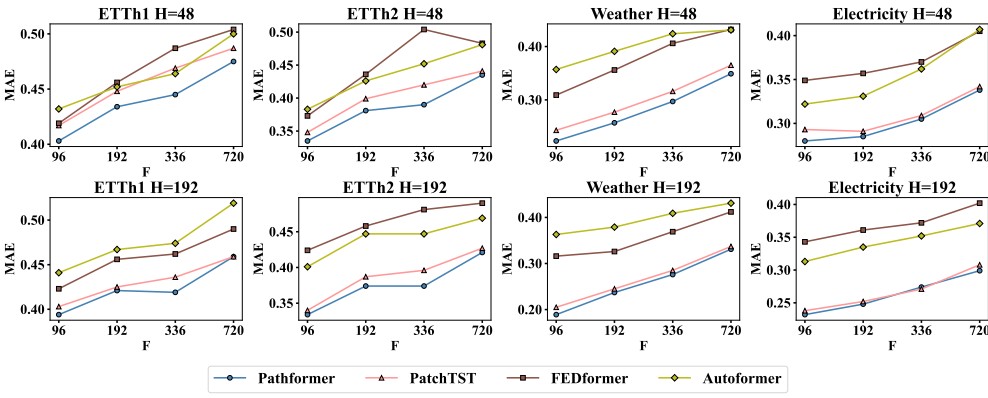

Figure 5: Results with different input length for ETTh1, ETTh2, Weather and Electricity.

Table 7: Multivariate time series forecasting results. The input length $H = 48$, and the prediction length $F \in \{96, 192, 336, 720\}$. The best results are highlighted in bold.

| Models | | Pathformer | | PatchTST | | FEDformer | | Autoformer | |
|---|---|---|---|---|---|---|---|---|---|
| Metric | | MSE | MAE | MSE | MAE | MSE | MAE | MSE | MAE |
| ETTh1 | 96 | 0.390 | **0.403** | 0.410 | 0.417 | **0.382** | 0.419 | 0.406 | 0.432 |
| | 192 | 0.454 | **0.434** | 0.469 | 0.448 | **0.451** | 0.456 | 0.451 | 0.452 |
| | 336 | **0.483** | **0.445** | 0.516 | 0.469 | 0.499 | 0.487 | 0.461 | 0.464 |
| | 720 | **0.507** | **0.475** | 0.509 | 0.487 | 0.510 | 0.504 | 0.498 | 0.500 |
| ETTh2 | 96 | **0.295** | **0.335** | 0.307 | 0.348 | 0.330 | 0.373 | 0.344 | 0.383 |
| | 192 | **0.366** | **0.381** | 0.397 | 0.399 | 0.440 | 0.436 | 0.425 | 0.426 |
| | 336 | **0.368** | **0.390** | 0.412 | 0.420 | 0.543 | 0.504 | 0.445 | 0.452 |
| | 720 | **0.428** | **0.435** | 0.434 | 0.441 | 0.471 | 0.483 | 0.483 | 0.481 |
| ETTm1 | 96 | **0.420** | **0.392** | 0.424 | 0.403 | 0.428 | 0.432 | 0.745 | 0.556 |
| | 192 | **0.446** | **0.410** | 0.468 | 0.429 | 0.476 | 0.460 | 0.715 | 0.556 |
| | 336 | **0.469** | **0.431** | 0.501 | 0.453 | 0.526 | 0.494 | 0.816 | 0.590 |
| | 720 | **0.512** | **0.465** | 0.553 | 0.484 | 0.630 | 0.528 | 0.746 | 0.572 |
| ETTm2 | 96 | **0.181** | **0.256** | 0.189 | 0.272 | 0.185 | 0.274 | 0.211 | 0.299 |
| | 192 | **0.251** | **0.301** | 0.260 | 0.371 | 0.256 | 0.318 | 0.277 | 0.388 |
| | 336 | **0.323** | **0.349** | 0.328 | 0.359 | 0.329 | 0.365 | 0.347 | 0.380 |
| | 720 | **0.420** | **0.406** | 0.429 | 0.415 | 0.447 | 0.432 | 0.441 | 0.432 |
| Weather | 96 | **0.188** | **0.223** | 0.212 | 0.243 | 0.241 | 0.309 | 0.291 | 0.357 |
| | 192 | **0.227** | **0.257** | 0.254 | 0.277 | 0.308 | 0.356 | 0.349 | 0.391 |
| | 336 | **0.276** | **0.297** | 0.310 | 0.316 | 0.385 | 0.406 | 0.409 | 0.424 |
| | 720 | **0.345** | **0.349** | 0.385 | 0.365 | 0.438 | 0.432 | 0.437 | 0.431 |
| Electricity | 96 | **0.201** | **0.280** | 0.225 | 0.293 | 0.240 | 0.349 | 0.211 | 0.322 |
| | 192 | **0.210** | **0.285** | 0.229 | 0.299 | 0.248 | 0.357 | 0.224 | 0.331 |
| | 336 | **0.236** | **0.305** | 0.239 | 0.316 | 0.265 | 0.370 | 0.259 | 0.362 |
| | 720 | **0.272** | **0.338** | 0.282 | 0.349 | 0.326 | 0.405 | 0.313 | 0.407 |

Table 8: Multivariate time series forecasting results. The input length $H = 192$, and the prediction length $F \in \{96, 192, 336, 720\}$. The best results are highlighted in bold.

| Models | | Pathformer | | PatchTST | | FEDformer | | Autoformer | |
|---|---|---|---|---|---|---|---|---|---|
| Metric | | MSE | MAE | MSE | MAE | MSE | MAE | MSE | MAE |
| ETTh1 | 96 | **0.377** | **0.394** | 0.384 | 0.403 | 0.388 | 0.423 | 0.430 | 0.441 |
| | 192 | **0.428** | **0.421** | 0.428 | 0.425 | 0.433 | 0.456 | 0.487 | 0.467 |
| | 336 | **0.424** | **0.419** | 0.452 | 0.436 | 0.445 | 0.462 | 0.478 | 0.474 |
| | 720 | 0.474 | **0.459** | **0.453** | 0.459 | 0.476 | 0.490 | 0.518 | 0.519 |
| ETTh2 | 96 | **0.283** | **0.334** | 0.285 | 0.340 | 0.397 | 0.424 | 0.362 | 0.401 |
| | 192 | **0.343** | **0.374** | 0.356 | 0.387 | 0.439 | 0.458 | 0.430 | 0.447 |
| | 336 | **0.332** | **0.374** | 0.351 | 0.396 | 0.471 | 0.481 | 0.408 | 0.447 |
| | 720 | **0.393** | **0.421** | 0.395 | 0.427 | 0.479 | 0.490 | 0.440 | 0.469 |
| ETTm1 | 96 | **0.295** | **0.335** | 0.295 | 0.345 | 0.381 | 0.424 | 0.510 | 0.428 |
| | 192 | 0.336 | **0.361** | **0.330** | 0.365 | 0.412 | 0.441 | 0.619 | 0.545 |
| | 336 | **0.359** | **0.384** | 0.364 | 0.388 | 0.435 | 0.455 | 0.561 | 0.500 |
| | 720 | 0.432 | **0.420** | **0.423** | 0.424 | 0.473 | 0.474 | 0.580 | 0.512 |
| ETTm2 | 96 | **0.169** | **0.250** | 0.169 | 0.254 | 0.223 | 0.305 | 0.244 | 0.321 |
| | 192 | **0.230** | **0.290** | 0.230 | 0.294 | 0.281 | 0.339 | 0.302 | 0.362 |
| | 336 | 0.286 | **0.328** | **0.281** | 0.329 | 0.321 | 0.364 | 0.346 | 0.390 |
| | 720 | 0.375 | **0.384** | 0.373 | 0.384 | 0.417 | 0.420 | 0.423 | 0.428 |
| Weather | 96 | **0.152** | **0.189** | 0.160 | 0.205 | 0.239 | 0.316 | 0.298 | 0.363 |
| | 192 | **0.198** | **0.237** | 0.204 | 0.245 | 0.274 | 0.326 | 0.322 | 0.379 |
| | 336 | **0.246** | **0.276** | 0.258 | 0.285 | 0.334 | 0.369 | 0.378 | 0.409 |
| | 720 | **0.329** | **0.331** | 0.329 | 0.337 | 0.401 | 0.412 | 0.435 | 0.431 |
| Electricity | 96 | **0.136** | **0.232** | 0.146 | 0.240 | 0.231 | 0.343 | 0.198 | 0.313 |
| | 192 | **0.143** | **0.248** | 0.152 | 0.252 | 0.258 | 0.361 | 0.218 | 0.335 |
| | 336 | **0.172** | **0.274** | 0.178 | 0.271 | 0.273 | 0.372 | 0.252 | 0.352 |
| | 720 | **0.218** | **0.299** | 0.223 | 0.308 | 0.308 | 0.402 | 0.275 | 0.371 |

## A.4    MORE COMPARISONS WITH SOME BASIC BASELINES

To validate the effectiveness of Pathformer, we conducted extensive experiments with some recent basic baselines that exhibited good performance: DLinear, NLinear, and N-HiTS, using long input sequence length ($H = 336$). As depicted in Table 9, our proposed model Pathformer outperforms

Table 9: Multivariate time series forecasting results. The input length $H = 336$ ( for ILI dataset $H = 106$ ), and the prediction length $F \in \{96, 192, 336, 720\}$ ( for ILI dataset $F \in \{24, 36, 48, 60\}$ ). The best results are highlighted in bold.

| Method | | Pathformer | | DLinear | | NLinear | | N-HiTS | |
|---|---|---|---|---|---|---|---|---|---|
| Metric | | MSE | MAE | MSE | MAE | MSE | MAE | MSE | MAE |
| ETTh1 | 96 | **0.369** | **0.395** | 0.375 | 0.399 | 0.374 | 0.394 | 0.378 | 0.393 |
| | 192 | 0.414 | 0.418 | **0.405** | 0.416 | 0.408 | **0.415** | 0.427 | 0.436 |
| | 336 | **0.401** | **0.419** | 0.439 | 0.443 | 0.429 | 0.427 | 0.458 | 0.484 |
| | 720 | **0.440** | **0.452** | 0.472 | 0.490 | 0.440 | 0.453 | 0.561 | 0.501 |
| ETTh2 | 96 | 0.276 | **0.334** | 0.289 | 0.353 | 0.277 | 0.338 | **0.274** | 0.345 |
| | 192 | **0.329** | **0.372** | 0.383 | 0.418 | 0.344 | 0.381 | 0.353 | 0.401 |
| | 336 | **0.324** | **0.377** | 0.448 | 0.465 | 0.357 | 0.400 | 0.382 | 0.425 |
| | 720 | **0.366** | **0.410** | 0.605 | 0.551 | 0.394 | 0.436 | 0.625 | 0.557 |
| ETTm1 | 96 | **0.285** | **0.336** | 0.299 | 0.353 | 0.306 | 0.348 | 0.302 | 0.350 |
| | 192 | **0.331** | **0.361** | 0.335 | 0.365 | 0.349 | 0.375 | 0.347 | 0.383 |
| | 336 | **0.362** | **0.382** | 0.369 | 0.386 | 0.375 | 0.388 | 0.369 | 0.402 |
| | 720 | **0.412** | **0.414** | 0.425 | 0.421 | 0.433 | 0.422 | 0.431 | 0.441 |
| ETTm2 | 96 | **0.163** | **0.248** | 0.167 | 0.260 | 0.167 | 0.255 | 0.176 | 0.255 |
| | 192 | **0.220** | **0.286** | 0.224 | 0.303 | 0.221 | 0.293 | 0.245 | 0.305 |
| | 336 | 0.275 | **0.325** | 0.281 | 0.342 | **0.274** | 0.327 | 0.295 | 0.346 |
| | 720 | **0.363** | **0.381** | 0.397 | 0.421 | 0.368 | 0.384 | 0.401 | 0.413 |
| Weather | 96 | **0.144** | **0.184** | 0.176 | 0.237 | 0.182 | 0.232 | 0.158 | 0.195 |
| | 192 | **0.191** | **0.229** | 0.220 | 0.282 | 0.225 | 0.269 | 0.211 | 0.247 |
| | 336 | **0.234** | **0.268** | 0.265 | 0.319 | 0.271 | 0.301 | 0.274 | 0.300 |
| | 720 | **0.316** | **0.323** | 0.323 | 0.362 | 0.338 | 0.348 | 0.351 | 0.353 |
| Electricity | 96 | **0.134** | **0.218** | 0.140 | 0.237 | 0.141 | 0.237 | 0.147 | 0.249 |
| | 192 | **0.142** | **0.235** | 0.153 | 0.249 | 0.154 | 0.248 | 0.167 | 0.269 |
| | 336 | **0.162** | **0.257** | 0.169 | 0.267 | 0.171 | 0.265 | 0.186 | 0.290 |
| | 720 | **0.200** | **0.290** | 0.203 | 0.301 | 0.210 | 0.297 | 0.243 | 0.340 |
| ILI | 24 | **1.411** | **0.705** | 2.215 | 1.081 | 1.683 | 0.868 | 1.862 | 0.869 |
| | 36 | **1.365** | **0.727** | 1.963 | 0.963 | 1.703 | 0.859 | 2.071 | 0.934 |
| | 48 | **1.537** | **0.764** | 2.130 | 1.024 | 1.719 | 0.884 | 2.134 | 0.932 |
| | 60 | **1.418** | **0.772** | 2.368 | 1.096 | 1.819 | 0.917 | 2.137 | 1.968 |
| Traffic | 96 | **0.373** | **0.241** | 0.410 | 0.282 | 0.410 | 0.279 | 0.402 | 0.282 |
| | 192 | **0.380** | **0.252** | 0.423 | 0.287 | 0.423 | 0.284 | 0.420 | 0.297 |
| | 336 | **0.395** | **0.256** | 0.436 | 0.296 | 0.435 | 0.290 | 0.448 | 0.313 |
| | 720 | **0.425** | **0.280** | 0.466 | 0.315 | 0.464 | 0.307 | 0.539 | 0.353 |

these baselines for the input length 336. Zeng et al. (2023) point out that the previous Transformer cannot extract temporal relations well from longer input sequences, but our proposed Pathformer performs better with a longer input length, indicating that considering adaptive multi-scale modeling can be an effective way to enhance such a relation extraction ability of Transformers.

## A.5 DISCUSSION

### A.5.1 COMPARE WITH PATCHTST

PatchTST divides time series into patches, with empirical evidence proving that patching is an effective method to enhance model performance in time series forecasting. Our proposed model Pathformer extends the patching approach to incorporate multi-scale modeling. The main differences with PatchTST are as follows: (1) **Partitioning with Multiple Patch Sizes**: PatchTST employs a single patch size to partition time series, obtaining features with a singular resolution. In contrast, Pathformer utilizes multiple different patch sizes at each layer for partitioning. This approach captures multi-scale features from the perspective of temporal resolutions. (2) **Global correlations between patches and local details in each patch**: PatchTST performs attention between divided patches, overlooking the internal details in each patch. In contrast, Pathformer not only considers the correlations between patches but also the detailed information within each patch. It introduces dual attention(inter-patch attention and intra-patch attention) to integrate global correlations and local details, capturing multi-scale features from the perspective of temporal distances. (3)**Adaptive Multi-scale Modeling**: PatchTST employs a fixed patch size for all data, hindering the grasp of critical patterns in different time series. We propose adaptive pathways that dynamically select varying patch sizes tailored to the features of individual samples, enabling adaptive multi-scale modeling.

### A.5.2 COMPARE WITH N-HITS

N-HiTS utilizes the modeling of multi-scale features for time series forecasting, but it differs from Pathformer in the following aspects: (1) N-HiTS models time series features of different resolutions through multi-rate data sampling and hierarchical interpolation. In contrast, Pathformer not only takes into account time series features of different resolutions but also approaches multi-scale modeling from the perspective of temporal distance. Simultaneously considering temporal resolutions and temporal distances enables a more comprehensive approach to multi-scale modeling. (2) N-HiTS employs fixed sampling rates for multi-rate data sampling, lacking the ability to adaptively perform multi-scale modeling based on differences in time series samples. In contrast, Pathformer has the capability for adaptive multi-scale modeling. (3) N-HiTS adopts a linear structure to build its model framework, whereas Pathformer enables multi-scale modeling in a Transformer architecture.

### A.5.3 COMPARE WITH SCALEFORMER

Scaleformer also utilizes the modeling of multi-scale features for time series forecasting. It differs from Pathformer in the following aspects: (1) Scaleformer obtains multi-scale features with different temporal resolutions through downsampling. In contrast, Pathformer not only considers time series features of different resolutions but also models from the perspective of temporal distance, taking into account global correlations and local details. This provides a more comprehensive approach to multi-scale modeling through both temporal resolutions and temporal distances. (2) Scaleformer requires the allocation of a predictive model at different temporal resolutions, resulting in higher model complexity than Pathformer. (3) Scaleformer employs fixed sampling rates, while Pathformer has the capability for adaptive multi-scale modeling based on the differences in time series samples.

### A.6 EXPERIMENTS ON LARGE DATASETS

The current time series forecasting benchmarks are relatively small, and there is a concern that the predictive performance of the model might be influenced by overfitting. To address this issue, we explore larger datasets to validate the effectiveness of the proposed model. The detailed process is as follows: We seek larger datasets from two perspectives: data volume and the number of variables. We add two datasets, the Wind Power dataset, and the PEMS07 dataset, to evaluate the performance of Pathformer on larger datasets. The Wind Power dataset comprises 7397147 timestamps, reaching a sample size in the millions, and the PEMS07 dataset includes 883 variables. As depicted in Table 10, Pathformer demonstrates superior predictive performance on these larger datasets compared with some state-of-the-art methods such as PatchTST, DLinear, and Scaleformer.

Table 10: Results on large datasets: PEMS07 and Wind Power.

| Methods | | Pathformer | | PatchTST | | DLinear | | Scaleformer | |
|---|---|---|---|---|---|---|---|---|---|
| Metric | | MSE | MAE | MSE | MAE | MSE | MAE | MSE | MAE |
| PEMS07 | 96 | **0.135** | **0.243** | 0.146 | 0.259 | 0.564 | 0.536 | 0.152 | 0.268 |
| | 192 | **0.177** | **0.271** | 0.185 | 0.286 | 0.596 | 0.555 | 0.195 | 0.302 |
| | 336 | **0.188** | **0.278** | 0.205 | 0.289 | 0.475 | 0.482 | 0.276 | 0.394 |
| | 720 | **0.208** | **0.296** | 0.235 | 0.325 | 0.543 | 0.523 | 0.305 | 0.410 |
| Wind Power | 96 | **0.062** | **0.146** | 0.070 | 0.158 | 0.078 | 0.184 | 0.089 | 0.167 |
| | 192 | **0.123** | **0.214** | 0.131 | 0.237 | 0.133 | 0.252 | 0.163 | 0.246 |
| | 336 | **0.200** | **0.283** | 0.215 | 0.307 | 0.205 | 0.325 | 0.225 | 0.352 |
| | 720 | **0.388** | **0.414** | 0.404 | 0.429 | 0.407 | 0.457 | 0.414 | 0.426 |

### A.7 VISUALIZATION

We visualize the prediction results of Pathformer on the Electricity dataset. As illustrated in Figure 6, for prediction lengths $F = 96, 192, 336, 720$, the prediction curve closely aligns with the Ground Truth curve, indicating the outstanding predictive performance of Pathformer. Meanwhile, Pathformer demonstrates effectiveness in capturing multi-period and complex trends present in diverse samples. This serves as evidence of its adaptive modeling capability for multi-scale characteristics.

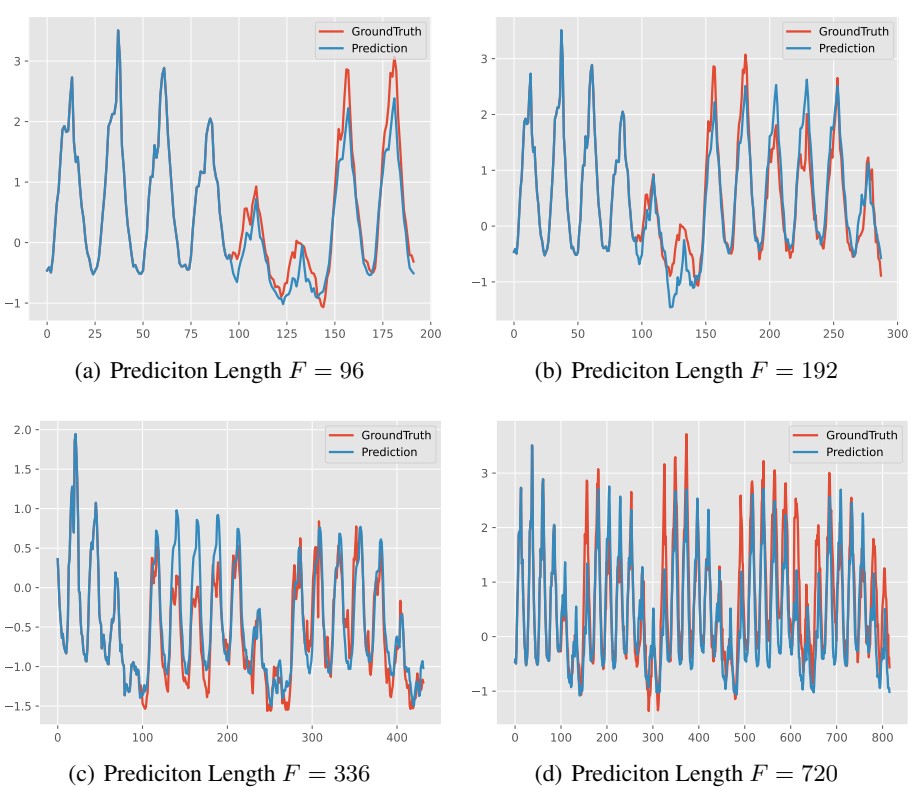

(a) Prediciton Length $F = 96$

(b) Prediciton Length $F = 192$

(c) Prediciton Length $F = 336$

(d) Prediciton Length $F = 720$

Figure 6: Visualization of Pathformer's prediction results on Electricity. The input length $H = 96$

