# OpenReview forum: "Pathformer: Multi-scale Transformers with Adaptive Pathways for Time Series Forecasting"
_ICLR.cc/2024/Conference — ICLR 2024 poster_

### Official Review · Reviewer_LNPN · 2023-11-04

**Soundness:** 3 good
**Presentation:** 2 fair
**Contribution:** 2 fair
**Rating:** 6
**Confidence:** 4

**Summary:**

This paper proposes multi-scale transformers with adaptive pathways. The proposed method integrates both temporal resolution and temporal distance for multi-scale modeling. It further enriches the multi-scale transformer with adaptive pathways. Experimental results showed the efficacy of proposed method and state-of-the-art performance.

**Strengths:**

1)  It's novel to propose multi-scale transformers with adaptive pathways.

2) It's novel to integrate both temporal resolution and temporal distance for multi-scale modeling.

3) The experiments showed state-of-the-art performance.

**Weaknesses:**

1.The current time series forecasting datasets are pretty small, and performance may be satuated or over-fitting. Could the method be used for larger datasets?

2. Scalformer [1] also uses the multi-scale nature of time series data, this paper didn't mention and compare the similarities and differences with Scalformer.

[1] Shabani, Amin, et al. "Scaleformer: iterative multi-scale refining transformers for time series forecasting." ICLR (2023).

**Questions:**

1. The current time series forecasting datasets are pretty small, and performance may be satuated or over-fitting. Could the method be used for larger datasets?

2. Scalformer [1] also uses hierarchical design and the scales of time series data, Could this paper compare the similarities and differences with Scalformer.

[1] Shabani, Amin, et al. "Scaleformer: iterative multi-scale refining transformers for time series forecasting." ICLR (2023).

---

> ### Author Response · Authors · 2023-11-18
> **Response to Reviewer LNPN**
>
> We would like to sincerely thank Reviewer LNPN for acknowledging our technical novelty and empirical contributions, as well as the comments regarding larger datasets and the related multi-scale baseline method. We have revised our paper accordingly.
>
> **Q1:** Results on larger datasets.
>
> **A1:** We seek larger datasets from two perspectives: data volume and the number of variables. We add two datasets, the Wind Power dataset and the PEMS07 dataset, to evaluate the performance of Pathformer on larger datasets. The Wind Power dataset comprises 7397147 timestamps, reaching a sample size in the millions, and the PEMS07 dataset includes 883 variables. If the reviewer may mention other larger datasets, we are also willing to test on them. Pathformer also demonstrates superior predictive performance on these larger datasets compared with some state-of-the-art methods such as PatchTST, DLinear, and Scaleformer. We add experiments on these larger datasets in $\underline{\text{Section A.6 of the revised supplementary}}$.
>
> | Methods |     | Pathformer        | PatchTST |     DLinear |     Scaleformer |
> |:---------:|:----------------:|:-------------:|:-------------------:|:-------------:|:-----:|
> | Metrics |     | MSE    / MAE   | MSE   /  MAE   | MSE   / MAE   | MSE   / MAE |
> | PEMS07    | 96  | **0.135**    / **0.243** | 0.146  / 0.259 | 0.564  / 0.536 |       0.152      /  0.268  |
> |         | 192 | **0.177**    / **0.271** | 0.185   / 0.286 | 0.596  / 0.555 |        0.195     /   0.302  |
> |         | 336 | **0.188**     / **0.278** | 0.205   / 0.289 | 0.475  / 0.482 |      0.276       /  0.394  |
> |         | 720 | **0.208**     / **0.296** | 0.235    / 0.325 | 0.543  / 0.523 |        0.305     /   0.410  |
> | Wind Power   | 96  | **0.062**     / **0.146** | 0.070    / 0.158 | 0.078   / 0.184 |      0.089       /  0.167   |
> |         | 192 | **0.123**     / **0.214** | 0.131    / 0.237 | 0.133  / 0.252 |      0.163       /  0.246   |
> |         | 336 | **0.200**       / **0.283** | 0.215   / 0.307 | 0.205  / 0.325 |      0.225       /    0.352 |
> |         | 720 | **0.388**     / **0.414** | 0.404   / 0.429 | 0.407  / 0.457 | 0.414 / 0.426 |
>
>
>
> **Q2:** Compare with Scaleformer.
>
> **A2:** We mentioned the method of Scaleformer in $\underline{\text{the related work of the original submission}}$. In the revision, we add Scaleformer results in $\underline{\text{Tabel 1 of the revised paper}}$ as an important baseline of multi-scale models. We also provide a more detailed comparison between Scaleformer and our model in $\underline{\text{the related work of the revised paper and Section A.5.3 of the revised supplementary}}$, as follows:
>
> Scaleformer also utilizes the modeling of multi-scale features for time series forecasting. It differs from our proposed Pathformer in the following aspects:
>
> - Scaleformer employs fixed sampling rates, while Pathformer has the capability for adaptive multi-scale modeling based on the differences in time series samples.
> - Scaleformer obtains multi-scale features with different temporal resolutions through downsampling. In contrast, Pathformer not only considers time series features of different resolutions but also models from the perspective of temporal distance, taking into account global correlations and local details. This provides a more comprehensive approach to multi-scale modeling through both temporal resolutions and temporal distances.
> - Scaleformer requires the allocation of a predictive model at different temporal resolutions, resulting in higher model complexity than Pathformer.

---

> ### Author Response · Authors · 2023-11-20
> **Looking forward to your feedback**
>
> Dear Reviewer LNPN,
>
> We would like to sincerely thank you for your time and efforts in reviewing our paper.
>
> We have made an extensive effort to try to successfully address your concerns, by conducting experiments on larger datasets with more timestamps and variables, comparing our proposed Pathformer with Scaleformer, and making revisions to the paper and appendix accordingly.
>
> We hope our response can effectively address your concerns, If you have any further concerns or questions, please do not hesitate to let us know, and we will respond timely.
>
> All the best,
>
> Authors

---

> > ### Author Response · Authors · 2023-11-21
> > **Response to Reviewer LNPN (before the end of rebuttal)**
> >
> > Dear Reviewer LNPN,
> >
> > Since the End of author/reviewer discussions is just in one day, may we know if our response addresses your main concerns? If so, we kindly ask for your reconsideration of the score. Should you have any further advice on the paper and/or our rebuttal, please let us know and we will be more than happy to engage in more discussion and paper improvements.
> >
> > Thank you so much for devoting time to improving our work!

---

### Official Review · Reviewer_Sqch · 2023-11-04

**Soundness:** 3 good
**Presentation:** 3 good
**Contribution:** 3 good
**Rating:** 6
**Confidence:** 4

**Summary:**

This paper presents a new multi-scale Transformer architecture for long-range time series modeling.The Discriminant Fourier Transform (DFT) is utilized to determine the patch sizes so as to divide the input time series into patches of different sizes, thus enabling cross-scale information fusion. In the multiscale transformer block, intra-patch attention and inter-patch attention mechanisms are utilized to perform attentional operations, thus enhancing the processing of temporal information.Experiments show the proposed method achieves state-of-the-art performance among existing models and exhibits superior generalization capabilities across different transfer scenarios.

**Strengths:**

1.The paper is well written and well-motivated.
2.The Multi-Scale Router combines the advantages of patch division and seasonality decomposition.
3.Dual attention helps to harmonize operations between intra-patch and inter-patch components, allowing the transformer to efficiently process time series data.

**Weaknesses:**

Time-series Dense Encoder (TiDE) is also a popular long-term time-series forecasting benchmark.But the paper does not include experiments comparing the proposed method to TiDE.

**Questions:**

Why is there a gap between the benchmark data in the paper and the original paper.

---

> ### Author Response · Authors · 2023-11-18
> **Response to Reviewer Sqch**
>
> We would like to sincerely thank Reviewer Sqch for acknowledging our technical contributions, providing the recent advanced method to compare with, and the suggestion for a more complete benchmark. We have revised our paper accordingly.
>
> **Q1:** Compare with Time-series Dense Encoder (TiDE).
>
> **A1:** The results of TiDE presented in the original paper are the best ones selected from different input sequence lengths of (48,96,192,336,720). Considering the urgency and fairness, we conduct a comparison under the condition of a fixed input sequence length of 96. We show results of some datasets here, and the complete results of other datasets and methods are available in $\underline{\text{Table 1 of the revised paper}}$.
>
>
> | Methods |  | Pathformer | TIDE |
> | :-------:|:-------:|:----------:|:---------:|
> |Metrics|  |MSE / MAE| MSE / MAE |
> | ETTm2      | 96|  0.170 / 0.248 | 0.182 / 0.264 |
> |            | 192|    0.238 / 0.295 |0.256 / 0.323 |
> |            | 336|  0.293 / 0.331 | 0.313 / 0.354 |
> |            | 720|  0.390 / 0.389 | 0.419 / 0.410|
> | Electricity| 96|  0.145 / 0.236   |    0.194/ 0.277  |
> |            | 192|  0.167 / 0.258  |   0.193 / 0.280   |
> |            | 336|  0.186 / 0.275  |    0.206 / 0.296  |
> |            | 720|   0.231 / 0.309 |    0.242 / 0.328  |
> | Weather    | 96 |  0.156 / 0.192  |      0.202 / 0.261             |
> |            | 192|  0.206 / 0.240  |      0.242 / 0.298 |
> |            | 336|  0.254 / 0.282  |      0.287 / 0.335 |
> |            | 720|  0.340 / 0.336  |      0.351 / 0.386 |
>
>
>
> **Q2:** More datasets in the benchmark.
>
> **A2:** We add the ILI and Traffic datasets in the benchmark commonly used by previous papers such as PatchTST. The complete results of other compared methods can be found in $\underline{\text{Table 1 of the revised paper}}$.
>
>
> | Methods | |Pathformer      | PatchTST |
> |:------------:|:---------------:|:-------:|:---------:|
> | Metrics     |     | MSE      / MAE   | MSE     / MAE   |
> | ILI        | 24  | 1.587   / 0.758 | 1.724   / 0.843 |
> |            | 36  | 1.429    / 0.711 | 1.536   / 0.752 |
> |            | 48  | 1.505    / 0.742 | 1.821   / 0.832 |
> |            | 60  | 1.731    / 0.799 | 1.923   / 0.842 |
> | Traffic    | 96  | 0.479    / 0.283 | 0.492   / 0.324 |
> |            | 192 | 0.484    / 0.292 | 0.487   / 0.303 |
> |            | 336 | 0.503    / 0.299 | 0.505   / 0.317 |
> |            | 720 | 0.537    / 0.322 | 0.542   / 0.337 |
>
> We also evaluate the Exchange_rate dataset, which is proposed in the benchmark of the Autoformer paper. Our proposed Pathformer also outperforms Autoformer on this dataset.
>
> | Methods | |Pathformer      | Autoformer|
> |:------------:|:---------------:|:-------:|:------:|
> | Metrics     |     | MSE      / MAE   | MSE     / MAE   |
> | Exchange    | 96  | 0.084    / 0.203 | 0.197   / 0.323 |
> |            | 192 | 0.178    / 0.300 | 0.300   / 0.369 |
> |            | 336 | 0.346    / 0.425 | 0.509   / 0.524 |
> |            | 720 | 0.889    / 0.704 | 1.447   / 0.941 |

---

> ### Author Response · Authors · 2023-11-20
> **Looking forward to your feedback**
>
> Dear Reviewer Sqch,
>
> We would like to sincerely thank you for your time and efforts in reviewing our paper.
>
> We have exerted significant effort to effectively address your concerns, by including experiments comparing the proposed Pathformer to TiDE, adding ILI, Traffic and Exchange datasets in the benchmark, and making revisions to the paper and appendix accordingly.
>
> We hope our response can address your concerns. If you have any further concerns or questions, please do not hesitate to let us know, and we will respond timely.
>
> All the best,
>
> Authors

---

### Official Review · Reviewer_5cD6 · 2023-11-07

**Soundness:** 3 good
**Presentation:** 3 good
**Contribution:** 3 good
**Rating:** 8
**Confidence:** 5

**Summary:**

The paper proposes a variation of PatchTST architecture in the context long-horizon time series forecating.

**Strengths:**

- An interesting study that makes an incremental step towards making transformers effective at long-horizon forecasting task
- Paper is clearly written and topic is important for the ICLR audience

**Weaknesses:**

- "Recent advances for time series forecasting are mainly based on Transformer architectures". I would say this statement is not aligned with the most recent empirical results and contradicts existing facts. Having read the following papers one could argue that the transformer based models in time series forecasting have been basically a disaster in the recent years, mainly because authors of papers based on transformer-driven models disregarded including some basic baselines in their studies. Please rewrite intro and related work accordingly.
    - Challu et al. N-HiTS: Neural hierarchical interpolation for time series forecasting. AAAI'23
    - Zeng et al. Are transformers effective for time series forecasting? AAAI'23
    - Li et al. Do Simpler Statistical Methods Perform Better in Multivariate Long Sequence Time-Series Forecasting? CIKM'23
- Not all datasets are present in the study. Please include additional results on ILI and Traffic from PatchTST
- Please include results from Zeng et al. Are transformers effective for time series forecasting? AAAI'23 in your table and you will see that your results are not state of the art. This makes the results unconvincing, because basically a very complex model is not able to use the same inputs as very simple models in an effective way. Transformer based papers have consistently failed to include appropriate baselines in the studies creating a large gap in methodology and undermining the ultimate reliability of these studies. The work can be interesting if authors show that with the proposed modifications a transformer based model can be more effective than much simpler and faster models presented in Zeng et al. and Li et al.
- The model seems to borrow conceptually very heavily from the PatchTST model without explicitly recognizing the source of inspiration. Without a detailed explanation of the actual difference between the two architectures the proposed architecture appears to be a minor perturbation of the original PatchTST.

**Questions:**

- When talking about multi-scale processing in related work please discuss relation to Challu et al. N-HiTS: Neural hierarchical interpolation for time series forecasting AAAI'23, which seems to be relevant work on multi-scale modelling

---

> ### Author Response · Authors · 2023-11-18
> **Response to Reviewer 5cD6 (Part I)**
>
> We would like to sincerely thank Reviewer 5cD6 for providing a detailed review and insightful comments regarding important basic baselines, more datasets, and more detailed comparisons with existing methods. We have revised our paper accordingly.
>
> **Q1:** More discussion on some basic baselines.
>
> **A1:** Thanks a lot for raising this valuable comment. We agree with the reviewer that we should also include some basic baselines as proposed in the comment, which may put our paper in a more correct context, give the readers a clearer view of the potentials and challenges of current Transformer methods, and make our contributions more convincing. We have made the following revisions according to your comment:
>
> - We add these important basic baselines in $\underline{\text{Introduction or Related Work in the revised paper}}$, and rewrite these two parts accordingly.
>
> - We compare the performance of our method with these baselines, such as DLinear, NLinear and NHITS, to make our empirical improvements more convincing. The detailed experimental results are in the following parts.
>
>
> **Q2:** Include additional results on ILI and Traffic from PatchTST
>
> **A2:** We add the ILI and Traffic datasets in $\underline{\text{Table 1 in the revised paper}}$. Here, we show some models with superior performance: PatchTST, DLinear, FEDformer, and Autoformer (bold indicates the best). Results of other compared methods are also included in the revised paper.
>
>
>
> | Methods      |     | PathFormer |     PatchTST  | DLinear |      Fedformer   |   Autoformer |
> |:-------------:|:----------:|:-------------------:|:----------------:|:-----------------:|:------------:|:-------------:|
> | Metrics     |     | MSE / MAE   | MSE / MAE   | MSE  / MAE   | MSE / MAE|   MSE / MAE |
> | ILI       | 96  | **1.587** / **0.758** | 1.724 / 0.843 | 2.573 / 1.073 | 2.624 / 1.095 | 2.906 / 1.182 |
> |             | 192 | **1.429** / **0.711** | 1.536 / 0.752 | 2.673 / 1.085 | 2.516 / 1.021 | 2.585 / 1.038 |
> |             | 336 | **1.505** / **0.742** | 1.821 / 0.832 | 2.773 / 1.126 | 2.505 / 1.041 | 3.024 / 1.145 |
> |             | 720 | **1.731** / **0.799** | 1.923 /  0.842 | 2.827 / 1.152 | 2.742 / 1.122 | 2.761 / 1.114 |
> | Traffic       | 96  | **0.479** / **0.283** | 0.492  / 0.324 | 0.648 / 0.396 | 0.576 / 0.359 | 0.597 / 0.371 |
> |             | 192 | **0.484** / **0.292** | 0.487  / 0.303 | 0.613 / 0.614 | 0.610 / 0.380 | 0.607 / 0.382 |
> |             | 336 | **0.503** / **0.299** | 0.505 / 0.317 | 0.614 / 0.383 | 0.608 / 0.375 | 0.623 / 0.387 |
> |             | 720 | **0.537** / **0.322** | 0.542  / 0.337 | 0.655 / 0.405 | 0.621 / 0.375 | 0.639 / 0.395 |

---

> ### Author Response · Authors · 2023-11-18
> **Response to Reviewer 5cD6 (Part II)**
>
> **Q3:** Include and compare with the results from Zeng et al.
>
> **A3:** Some results presented in the paper of Zeng et al. (DLinear) are better than ours in Table 1, which is because that DLinear uses a **larger input length** (H=336) than ours (H=96). To ensure a fair comparison, we conduct separate experiments for input sequence lengths (H) of 96 and 336. Here, we show results of some models and datasets, with the complete results of other models and on other datasets available in the $\underline{\text{Table 1 in the revised paper and Table 9 in supplementary}}$.
>
> **The results for the input sequence length H=96:**
>
> | Methods |     | Pathformer |    DLinear     | NLinear |
> |:---------:|:-----:|:------------------:|:---------------:|:----:|
> | Metrics |     | MSE       /MAE   | MSE     / MAE   |MSE / MAE|
> | ETTm1   | 96  | **0.316**      / **0.346** | 0.342   / 0.370 |0.339 / 0.369|
> |         | 192 | **0.366**      / **0.370** | 0.383   / 0.394 |0.379 / 0.386|
> |         | 336 | **0.386**     / **0.394** | 0.413   / 0.414 |0.411 / 0.407|
> |         | 720 | **0.460**    / **0.432** | 0.472   / 0.452 |0.478 / 0.442|
> | Weather | 96  | **0.156**      / **0.192** | 0.195   / 0.253 |0.168 / 0.208|
> |         | 192 | **0.206**     / **0.240** | 0.239   / 0.299 |0.217 / 0.255|
> |         | 336 | **0.254**     / **0.282**| 0.282   / 0.333 |0.267 / 0.292|
> |         | 720 | **0.340**    / **0.336** | 0.352   / 0.390 |0.351 / 0.346|
> | ILI     | 24  | **1.587**     / **0.758** | 2.573   / 1.073 |2.725 / 1.069|
> |         | 36  | **1.429**     / **0.711** | 2.673   / 1.085 |2.530 / 1.032|
> |         | 48  | **1.505**     / **0.742**| 2.773   / 1.126 |2.510 / 1.031|
> |         | 60  | **1.731**     / **0.799** | 2.827   / 1.152 |2.492 / 1.026|
>
> **The results for the input sequence length H=336:**
>
> | Methods  |     | Pathformer        | DLinear     | NLinear    |
> |:---------:|:-----:|:------------------:|:----------------:|:----------------:|
> | Metrics  |     | MSE       / MAE    | MSE     / MAE    | MSE     / MAE    | MSE   / MAE    |
> | ETTm1   | 96  | **0.285**    / **0.336**  | 0.299   / 0.353  | 0.306   / 0.348  |
> |         | 192 | **0.331**    / **0.361**  | 0.335   / 0.365  | 0.349   / 0.375  |
> |         | 336 | **0.362**     / **0.382**  | 0.369  / 0.386  | 0.375   / 0.388  |
> |         | 720 | **0.412**    / **0.414**  | 0.425   / 0.421  | 0.433   / 0.422  |
> | Weather | 96  | **0.144**   / **0.184**  | 0.176  / 0.237  | 0.182   / 0.232  |
> |         | 192 | **0.191**   / **0.229**  | 0.220 / 0.282  | 0.225  / 0.269  |
> |         | 336 | **0.234**   / **0.268**  | 0.265  / 0.319  | 0.271  / 0.301  |
> |         | 720 | **0.316**    / **0.323**  | 0.323   / 0.362  | 0.338    / 0.348  |
> | ILI     | 24  | **1.411**      / **0.705**  | 2.215   / 1.081  | 1.683    / 0.868  |
> |         | 36  | **1.365**      / **0.727**  | 1.963   / 0.963  | 1.703    / 0.859  |
> |         | 48  | **1.537**      / **0.764**  | 2.130   / 1.024  | 1.719    / 0.884  |
> |         | 60  | **1.418**      / **0.772**  | 2.368   / 1.096  | 1.819    / 0.917  |
>
> The results above reveal that our proposed model Pathformer outperforms DLinear with both input sequence lengths of 96 and 336. Zeng et al. point out that previous Transformer cannot extract temporal relations well from longer input sequences, but our proposed Pathformer performs better with a longer input length, indicating that considering adaptive multi-scale modeling can be an effective way to enhance such a relation extraction ability of Transformers.

---

> ### Author Response · Authors · 2023-11-18
> **Response to Reviewer 5cD6 (Part III)**
>
> We also conduct experiments to compare the **generalization capabilities** of Pathformer and DLinear, where we train the model on one dataset and test the performance on other datasets. The specific experimental setting is the same with Section 4.2 of the paper.
>
> **The results of the generalization on other datasets:**
>
> | Methods  |     | Pathformer  | DLinear     |
> |:----------:|:-----:|:-----------------:|:---------------:|
> | Metrics  |     | MSE        / MAE    | MSE      / MAE    | MSE     / MAE    |
> | ETTh2    | 96  | **0.340**      / **0.369**  |  0.37    / 0.398  |
> |          | 192 | **0.411**      / **0.406**  |  0.502   / 0.498  |
> |          | 336 | **0.384**      / **0.401**  |  0.563   / 0.531  |
> |          | 720 | **0.450**      / **0.448**  |  0.723   / 0.605  |
> | ETTm2    | 96  | **0.220**      / **0.294**  |  0.272   / 0.325  |
> |          | 192 | **0.258**      / **0.306**  |  0.352   / 0.398  |
> |          | 336 | **0.325**      / **0.350**  |  0.425   / 0.478  |
> |          | 720 | **0.422**      / **0.408**  |  0.553   / 0.517  |
> | Cluster-A | 24  | **0.121**     / **0.223**  |  0.342   / 0.418  |
> |          | 48  | **0.186**      / **0.281**  |  0.389   / 0.468  |
> |          | 96  | **0.249**      / **0.334**  |  0.392   / 0.473  |
> |          | 192 | **0.372**      / **0.416**  |  0.523   / 0.616  |
> | Cluster-B | 24  | **0.140**     / **0.243**  |  0.201   / 0.342  |
> |          | 48  | **0.202**      / **0.298**  |  0.256   / 0.387  |
> |          | 96  | **0.296**      / **0.357**  |  0.389   / 0.476  |
> |          | 192 | **0.464**      / **0.468**  |  0.628   / 0.635  |
> | Cluster-C | 24  | **0.069**      / **0.173**  |  0.145   / 0.242  |
> |          | 48  | **0.144**      / **0.254**  |  0.267   / 0.387  |
> |          | 96  | **0.174**     / **0.284**  |  0.411   / 0.512  |
> |          | 192 | **0.327**      / **0.386**  |  0.522   / 0.532  |
>
> In the generalization experiments, Pathformer still outperforms DLinear. This indicates that a relatively complex Transformer architecture may have better generalization capabilities than a simple linear model.
>
> **Q4:** Compare with PatchTST
>
> **A4:** We mention the source of inspiration of patching in $\underline{\text{Section 3 of our revised paper}}$. We also want to clarify that Pathformer extends from patch division to realize adaptive multi-scale modeling, which is a novel design. It is not a perturbation of PatchTST, as how the series are divided, and how the correlations are modeled are both designed differently. The main differences with PatchTST are as follows:
>
> - **Adaptive Multi-scale Modeling:**
> PatchTST employs a fixed patch size for all data, hindering the grasp of critical patterns in different time series. We are the first to propose adaptive pathways that dynamically select varying patch sizes tailored to the dynamic features of individual samples, enabling adaptive multi-scale modeling.
>
> - **Partitioning with Multiple Patch Sizes:**
> PatchTST employs a single patch size to partition time series, obtaining features with a singular resolution. Pathformer utilizes multiple different patch sizes for partitioning, which captures multi-scale features from various temporal resolutions.
>
> - **Global correlations between patches and local details in each patch:**
> PatchTST performs attention between divided patches, overlooking the internal details in each patch. Pathformer not only considers the correlations between patches but also the detailed information within each patch. It introduces dual attention (inter-patch attention and intra-patch attention) to integrate global correlations and local details, capturing multi-scale features from various temporal distances.
>
> We also add the above discussion in $\underline{\text{Section A.5.1 of the revised supplementary}}$.
>
> **Q5:** Compare with NHITS
>
> **A5:** NHITS also models multi-scale features for time series forecasting, and Pathformer differs from it in the following aspects:
>
> - NHITS models time series features of different resolutions through multi-rate data sampling and hierarchical interpolation. Pathformer not only takes into account time series features of different resolutions but also approaches multi-scale modeling from the perspective of temporal distance. Simultaneously considering temporal resolutions and temporal distances enables a more comprehensive approach to multi-scale modeling.
> - NHITS employs fixed sampling rates for multi-rate data sampling, lacking the ability to adaptively perform multi-scale modeling based on differences in time series samples. In contrast, Pathformer has the capability for adaptive multi-scale modeling.
> - NHITS adopts a linear structure to build its model framework, whereas Pathformer enables multi-scale modeling in a Transformer architecture.
>
> We add the above discussion in $\underline{\text{Section A.5.2 of the revised supplementary}}$. We also compare our performance with NHITS in $\underline{\text{Table 9 of the revised supplementary}}$

---

> ### Author Response · Authors · 2023-11-20
> **Looking forward to your feedback**
>
> Dear Reviewer 5cD6,
>
> We would like to express our sincere gratitude for your time and efforts in reviewing our paper.
>
> We have made an extensive effort to try to successfully address your concerns. In our response:
>
> - We provide more discussions with basic models, such as DLinear, NLinear, and rewrite the introduction and related work according to your suggestions.
>
> - We add ILI and Traffic datasets from PatchTST.
>
> - We compare our performance with these basic baselines on diverse datasets to make the results of the proposed pathformer more convincing.
>
> - We also compare Pathformer with PatchTST and NHITS to show the novelty and effectiveness of Pathformer and make revisions to the paper and appendix accordingly.
>
> We hope our response can address your concerns. If you have any further concerns or questions, please do not hesitate to inform us, and we will be more than happy to address them promptly.

---

> > ### Author Response · Authors · 2023-11-21
> > **Response to Reviewer 5cD6 (before the end of discussion)**
> >
> > Dear Reviewer 5cD6,
> >
> > Since the End of author/reviewer discussions is just in one day, may we know if our response addresses your main concerns? If so, we kindly ask for your reconsideration of the score.
> >
> > Should you have any further advice on the paper and/or our rebuttal, please let us know and we will be more than happy to engage in more discussion and paper improvements. We would really appreciate it if our next round of communication could leave time for us to resolve any of your remaining or new questions.
> >
> > Thank you so much for devoting time to improving our paper!

---

### Meta-Review · Area_Chair_2fwj · 2023-12-14

**Metareview:**

The paper introduces a novel approach called Pathformer, aiming to enhance time series forecasting using Transformer-based models. Traditional methods often struggle to model time series comprehensively across various scales. In response, the proposed Pathformer leverages multi-scale transformers with adaptive pathways to address this limitation. This paper represents a noteworthy advancement, introducing novelty by proposing multi-scale transformers with adaptive pathways and integrating both temporal resolution and temporal distance for comprehensive multi-scale modeling, ultimately achieving state-of-the-art results in long-horizon forecasting tasks. The paper's weaknesses include the need for a revised introduction reflecting recent empirical results, insufficient dataset inclusion and baseline comparisons, and unclear demonstration of the proposed modifications' superiority over simpler models. Conceptual similarities with PatchTST should be addressed, and scalability to larger datasets must be explored. Additionally, similarities with Scalformer and comparisons with TiDE are lacking. After the rebuttal, most of these problems have been addressed. Therefore, I recommend accepting this paper.

**Justification For Why Not Higher Score:**

This paper investigates the advantages of multiscale features for LSTF tasks. The concept of multiscale in LSTF has been explored in some prior works, so the novelty of this work is not particularly substantial.

**Justification For Why Not Lower Score:**

Although the application of multiscale features in the LSTF domain is not novel in this work, the proposed method significantly differs from previous approaches. With well-organized experiments and clear results, consensus among all reviewers has been reached, hence, recommending acceptance.

---

### Decision · Program_Chairs · 2024-01-16

Accept (poster)